# Random Forest Autoencoders for Guided Representation Learning

**Adrien Aumon**[1†]
adrien.aumon@umontreal.ca

**Shuang Ni**[1†]
shuang.ni@mila.quebec

**Myriam Lizotte**[1]
myriam.lizotte@mila.quebec

**Guy Wolf**[1]
guy.wolf@umontreal.ca

**Kevin R. Moon**[2]
kevin.moon@usu.edu

**Jake S. Rhodes**[3‡]
rhodes@stat.byu.edu

[1]Université de Montréal; Mila – Quebec AI Institute
[2]Utah State University [3]Brigham Young University
[†]Equal contribution [‡]Corresponding author

## Abstract

Extensive research has produced robust methods for unsupervised data visualization. Yet supervised visualization—where expert labels guide representations—remains underexplored, as most supervised approaches prioritize classification over visualization. Recently, RF-PHATE, a diffusion-based manifold learning method leveraging random forests and information geometry, marked significant progress in supervised visualization. However, its lack of an explicit mapping function limits scalability and its application to unseen data, posing challenges for large datasets and label-scarce scenarios. To overcome these limitations, we introduce Random Forest Autoencoders (RF-AE), a neural network-based framework for out-of-sample kernel extension that combines the flexibility of autoencoders with the supervised learning strengths of random forests and the geometry captured by RF-PHATE. RF-AE enables efficient out-of-sample supervised visualization and outperforms existing methods, including RF-PHATE's standard kernel extension, in both accuracy and interpretability. Additionally, RF-AE is modality-agnostic, demonstrates strong robustness to hyperparameters, supports both classification and regression, and natively accommodates missing feature values. Our code is available at https://github.com/JakeSRhodesLab/RF-AE.

## 1   Introduction

Manifold learning-based visualization methods, such as $t$-SNE [1], UMAP [2], and PHATE [3], are essential for exploring high-dimensional data by revealing patterns, clusters, and outliers through low-dimensional embeddings. While these methods excel at uncovering dominant data structures, they often fail to capture task-specific insights when auxiliary labels or metadata are available. Supervised approaches like RF-PHATE [4] bridge this gap by integrating label information into the kernel function through Random Forest-derived proximities [5], generating representations that align with domain-specific objectives without introducing the exaggerated separations or distortions seen in class-conditional methods [6]. Specifically, RF-PHATE has provided critical insights in biology,

39th Conference on Neural Information Processing Systems (NeurIPS 2025).

such as identifying multiple sclerosis subtypes, demonstrating antioxidant effects in lung cancer cells, and aligning COVID-19 antibody profiles with patient outcomes [4].

However, most supervised and unsupervised manifold learning algorithms generate fixed coordinates within a latent space but lack a mechanism to accommodate new observations. Therefore, to embed previously unseen data, the algorithm must rerun with the new data as part of the training set. One well-known solution to this lack of out-of-sample (OOS) support is the Nyström extension [7] and its variants, such as linear reconstruction [8] or geometric harmonics [9]. While influential, these approaches rely on linear kernel mappings and unconstrained least-squares minimization, making them highly sensitive to the quality of the training set and often inadequate for preserving complex manifold geometry in a way that generalizes effectively to new inputs [10, 11]. Recent neural network-based approaches, such as parametric variants of UMAP [12] or Geometry-Regularized Autoencoders (GRAE) [13, 14], offer promising alternatives for extending embeddings to OOS data points. Yet, most of these methods focus either on unsupervised structure or on predictive performance, without explicitly preserving label-informed geometry needed for interpretable supervised visualization.

In this study, we present Random Forest Autoencoders (RF-AE), an autoencoder (AE) architecture that addresses the underexplored setting of *supervised* OOS visualization, while taking inspiration from the principles of GRAE [14], which uses a manifold embedding to regularize the bottleneck layer. Instead of reconstructing the original input vectors, RF-AE incorporates supervised information by reconstructing Random Forest- Geometry- and Accuracy-Preserving (RF-GAP) proximities [5]. This framework induces an embedding function that simultaneously preserves local structure and class separability, and generalizes naturally to new data without requiring labels at inference. Unlike RF-PHATE [4], which provides supervised embeddings without a parametric mapping, RF-AE yields a generalizable and interpretable embedding function. Compared to existing neural network-based extensions, RF-AE incorporates a strong supervision signal tied to the relational structure between points. To further improve scalability, we also propose a prototype selection strategy that reduces memory and runtime demands during both training and inference.

Through extensive experiments, we show that RF-AE outperforms existing approaches in embedding new data while preserving the local and global structure of the important features for the underlying classification task. We also demonstrate that RF-AE's performance is robust to a wide range of hyperparameter configurations. RF-AE improves the adaptability and scalability of the manifold learning process, allowing for seamless integration of new data points while maintaining the desirable traits of established embedding methods.

## 2 Related work

### 2.1 Parametric embedding through multi-task autoencoders

Given a high-dimensional training dataset $X = \{\mathbf{x}_i \in \mathbb{R}^D \mid i = 1, \ldots, N\}$—where $X$ can represent tabular data, images, or other modalities—a manifold learning algorithm can be extended to test data by training a neural network, typically a multi-layer perceptron (MLP) to regress onto precomputed non-parametric embeddings $\mathbf{z}_i^G$, or by means of a cost function underlying the manifold learning algorithm, as in parametric $t$-SNE [15] and parametric UMAP [12]. However, solely training an MLP for this supervised task often leads to an under-constrained problem, resulting in solutions that memorize the data but fail to capture meaningful patterns or generalize effectively [16, 17]. Beyond implicit regularization techniques such as early stopping [18], dropout [19], or layer-wise pre-training [20], multi-task learning [21] has been shown to improve generalization. Early studies demonstrated that jointly learning tasks reduces the number of required samples [22, 23], while later work introduced trace-norm regularization on the weights of a linear, single hidden-layer neural network for a set of tasks [24, 25]. Motivated by Le et al. [26], who empirically showed that training neural networks to predict both target embeddings and inputs (reconstruction) improves generalization compared to encoder-only architectures, we focus on multi-task learning-based regularization in the context of regularized autoencoders.

AE networks are built of two parts: an encoder function $f(\mathbf{x}_i) = \mathbf{z}_i \in \mathbb{R}^d$ ($d \ll D$), which compresses the input data into a latent representation via a bottleneck layer [27], and a decoder function $g(\mathbf{z}_i) = \hat{\mathbf{x}}_i$, which maps the low-dimensional embedding back to the original input space. To ensure that $(g \circ f)(\mathbf{x}_i) = \hat{\mathbf{x}}_i \approx \mathbf{x}_i$, AEs minimize the average reconstruction loss

$L(f,g) = \frac{1}{N} \sum_{i=1}^{N} L_{recon}(\mathbf{x}_i, \hat{\mathbf{x}}_i)$ where $L_{recon}(\cdot, \cdot)$ is typically defined as the squared Euclidean distance. AEs learn compact data representations that relate meaningfully to the input data. However, standard AEs often fail to capture the intrinsic geometry of the data and do not produce interpretable embeddings [14]. This led authors to borrow principles of manifold learning to add geometrically motivated constraints to the latent space. Structural Deep Network Embedding (SDNE) [28] preserves both first- and second-order graph neighborhoods for graph-structured data by combining adjacency vector reconstruction with Laplacian Eigenmaps [29] regularization. Local and Global Graph Embedding Autoencoders [30] enforce two constraints on the embedding layer: a local constraint to cluster $k$-nearest neighbors and a global constraint to align data points with their class centers. VAE-SNE [31] integrates parametric $t$-SNE with variational AEs, enhancing global structure preservation while retaining $t$-SNE's strength in preserving local structure. GRAE [14] explicitly impose geometric consistency between the latent space and precomputed manifold embeddings. Other approaches focus on regularizing the decoder. Inspired by Denoising Autoencoders [32], Generalized Autoencoders [33] minimize the weighted mean squared error between the reconstruction $\hat{\mathbf{x}}$ and the $k$-nearest neighbors of the input $\mathbf{x}$, where weights reflect the predefined similarities between $\mathbf{x}$ and its neighbors. Centroid Encoders (CE) [34] minimize within-class reconstruction variance to ensure that same-class samples are mapped close to their respective centroids. Self-Supervised Network Projection (SSNP) [35] incorporates neighborhood information by jointly optimizing reconstruction and classification at the output layer, using existing labels or pseudo-labels generated through clustering. Neighborhood Reconstructing Autoencoders [36] extend reconstruction tasks by incorporating the neighbors of $\mathbf{x}$ alongside $\mathbf{x}$ itself, using a local quadratic approximation of the decoder at $f(\mathbf{x}) = \mathbf{z}$ to better capture the local geometry of the decoded manifold. Similarly, Geometric Autoencoders [37] introduce a regularization term in the reconstruction loss, leveraging the generalized Jacobian determinant computed at $f(\mathbf{x}) = \mathbf{z}$ to mitigate local contractions and distortions in latent representations. Still, most of these methods are unsupervised or apply supervision via class-conditional constraints, often leading to disrupted inter-class relationships [6, 38].

## 2.2 Kernel methods for OOS extension

Let $k(\cdot, \cdot)$ be a data-dependent symmetric positive definite kernel function $(\mathbf{x}, \mathbf{x}') \mapsto k(\mathbf{x}, \mathbf{x}') \geq 0$. For simplicity, we consider normalized kernel functions that satisfy the sum-to-one property $\sum_{i=1}^{N} k(\mathbf{x}, \mathbf{x}_i) = 1$. Kernel methods for OOS extensions seek an embedding function $\mathbf{k} \mapsto f(\mathbf{k}) = \mathbf{z} \in \mathbb{R}^d$ where the input $\mathbf{k} = \mathbf{k}_{\mathbf{x}} = [k(\mathbf{x}, \mathbf{x}_1) \quad \cdots \quad k(\mathbf{x}, \mathbf{x}_N)]$ is an $N$-dimensional similarity vector representing pairwise proximities between any instance $\mathbf{x}$ and all the points in the training set $X$. Under the linear assumption $f(\mathbf{k}) = \mathbf{k}\mathbf{W}$, where $\mathbf{W} \in \mathbb{R}^{N \times d}$ is a projection matrix to be determined, we directly define $\mathbf{W}$ in the context of a regression task [39, 40] by minimizing the least-squares error

$$\sum_{i=1}^{N} \|\mathbf{z}_i^G - f(\mathbf{k}_i)\|_2^2, \tag{1}$$

which yields the explicit solution $\mathbf{W} = \mathbf{K}^{-1}\mathbf{Y}$, where $\mathbf{K}^{-1}$ refers to the pseudo-inverse of the training Gram matrix $\mathbf{K} = [k(\mathbf{x}_i, \mathbf{x}_j)]_{1 \leq i,j \leq N}$, and $\mathbf{Y} = \begin{bmatrix} \mathbf{z}_1^G & \cdots & \mathbf{z}_N^G \end{bmatrix}^T$ contains the precomputed training manifold embeddings. In particular, for manifold learning algorithms that directly assign the low-dimensional coordinates from the eigenvectors of $\mathbf{K}$, e.g., Laplacian Eigenmaps [29], we have the well-known Nyström formula $\mathbf{W} = \mathbf{U}\mathbf{\Lambda}^{-1}$ [7, 41, 42], where $\mathbf{\Lambda}^{-1} = \text{diag}\left(\lambda_1^{-1}, \ldots, \lambda_d^{-1}\right)$. Here, $\lambda_i$ are the $d$ largest (or smallest, depending on the method) eigenvalues of $\mathbf{K}$, and $\mathbf{U}$ is the matrix whose columns are the corresponding eigenvectors. In Locally Linear Embedding [43] and PHATE [3], the authors suggested a default OOS extension through linear reconstruction $\mathbf{W} = \begin{bmatrix} \mathbf{z}_1^G & \cdots & \mathbf{z}_N^G \end{bmatrix}^T$. In diffusion-based methods, this provides an alternative means to compress the diffusion process through the training landmarks and has been shown to outperform a direct application of the Nyström extension to diffusion maps [44].

Unlike parametric extensions discussed in Section 2.1, kernel extensions learn an explicit embedding function using kernel representations rather than the original representations in the feature space. While kernel methods are powerful for handling high-dimensional datasets, they require computing pairwise kernels for all points in the training set, which can become computationally expensive for large datasets. In such cases, feature mappings offer greater scalability. However, kernel extensions have been shown to outperform direct parametric methods in unsupervised OOS visualization and classification tasks [40, 45]. Additionally, in supervised visualization, a carefully chosen kernel

mapping can effectively filter out irrelevant features, whereas feature mappings treat all features equally, potentially increasing sensitivity to noisy datasets. In particular, reconstructing RF-based proximities in an RF-PHATE-regularized AE produces OOS embeddings that better align with the ground-truth RF-PHATE embedding computed on the full data [46]. We further validated in Appendix B the ability of RF-GAP inputs to stabilize supervised manifold learning on a toy dataset (Appendix C). Therefore, we align our OOS extension framework with kernel extensions rather than direct parametric methods.

## 3 Methods

While traditional kernel extensions offer computational benefits over parametric methods, they are limited to linear kernel mappings and are primarily designed for unsupervised data visualization or classification using either unsupervised or class-conditional kernel functions [40, 45]. In this work, we expand the search space of the standard least-squares minimization problem in Eq. 1 to include general, potentially nonlinear kernel mapping functions $f$. We also propose a supervised kernel mapping based on Random Forests, specifically tailored for supervised data visualization. Building on the previous Section 2.1, we add a geometrically motivated regularizer to this regression task within a multi-task autoencoder framework.

In this section, we elaborate on the methodology related to our RF-AE framework to extend any kernel-based dimensionality reduction method with Random Forests and autoencoders for supervised data visualization. Specifically, we explain how we combined RF-GAP proximities and the visualization strengths of RF-PHATE with the flexibility of autoencoders to develop a new parametric supervised embedding method, Random Forest Autoencoders (RF-AE). Additionally, we introduce *structural importance alignment* as a metric for evaluating embedding models in supervised settings. Our code is available at https://github.com/JakeSRhodesLab/RF-AE.

### 3.1 Extended RF-GAP kernel function

The RF-GAP proximity [5] between (possibly unseen) instance $\mathbf{x}_i$ and training instance $\mathbf{x}_j$ is

$$p(\mathbf{x}_i, \mathbf{x}_j) = \frac{1}{|S_i|} \sum_{t \in S_i} \frac{c_j(t) \cdot I\left(j \in J_i(t)\right)}{|M_i(t)|},$$

where $S_i$ denotes the set of out-of-bag trees for observation $\mathbf{x}_i$, $c_j(t)$ is the number of in-bag repetitions for observation $\mathbf{x}_j$ in tree $t$, $I(\cdot)$ is the indicator function, $J_i(t)$ is the set of in-bag points residing in the terminal node of observation $\mathbf{x}_i$ in tree $t$, and $M_i(t)$ is the multiset of in-bag observation indices, including repetitions, co-occurring in a terminal node with $\mathbf{x}_i$ in tree $t$. Note that this definition naturally extends to OOS observations $\mathbf{x}_o \notin X$, which can be treated as out-of-bag for all trees. However, this definition requires that self-similarity be zero, that is, $p(\mathbf{x}_i, \mathbf{x}_i) = 0$. This is not suitable as a similarity measure in some applications. Due to the scale of the proximities—the rows sum to one [5], so the proximity values are all near zero for larger datasets—, it is not practical to re-assign self-similarities to one. Otherwise, self-similarity would carry equal weight to the combined significance of all other similarities. Instead, we assign values by, in essence, passing down an identical OOB point to all trees where the given observation is in-bag. That is, we define self-similarity as

$$p(\mathbf{x}_i, \mathbf{x}_i) = \frac{1}{|\bar{S}_i|} \sum_{t \in \bar{S}_i} \frac{c_i(t)}{|M_i(t)|},$$

where $\bar{S}_i$ is the set of trees for which $\mathbf{x}_i$ is in-bag. Under this formulation, $p(\mathbf{x}_i, \mathbf{x}_i)$ is on a scale more similar to other proximity values, and Proposition A.1 (Appendix A) guarantees that, on average, $p(\mathbf{x}_i, \mathbf{x}_i) > p(\mathbf{x}_i, \mathbf{x}_j)$. Now, we define the row-normalized RF-GAP similarity between a pair of training instances $\mathbf{x}_i$ and $\mathbf{x}_j$ as

$$\tilde{p}(\mathbf{x}_i, \mathbf{x}_j) = \frac{p(\mathbf{x}_i, \mathbf{x}_j)}{\sum_{j=1}^{N} p(\mathbf{x}_i, \mathbf{x}_j)} \tag{2}$$

We intentionally applied row-normalization to restore the sum-to-one property and refocus on the underlying geometry rather than sample distribution.

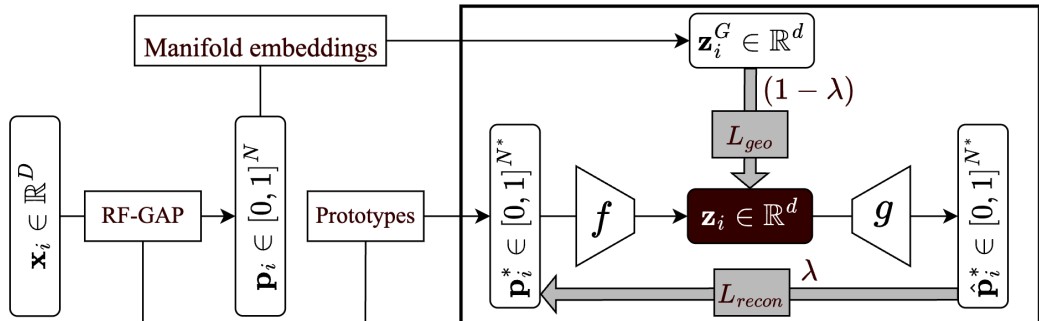

Figure 1: RF-AE architecture with prototype selection and geometric regularization. First, the original feature vectors $\mathbf{x}_i$ are transformed into one-step transition probability vectors $\mathbf{p}_i$ derived from RF-GAP proximities (Section 3.1). They are further reduced into lower-dimensional vectors $\mathbf{p}_i^*$ that represent transition probabilities to $N^* \ll N$ selected prototypes (Section 3.3). Meanwhile, manifold embeddings $\mathbf{z}_i^G$ are generated using RF-PHATE from the $\mathbf{p}_i$. Finally, $\mathbf{p}_i^*$ and $\mathbf{z}_i^G$ serve as input to the network within the enclosing box, training an encoder $f$ and a decoder $g$ by simultaneously minimizing the reconstruction loss $L_{recon}$ and the geometric loss $L_{geo}$ defined in Section 3.2.

## 3.2   RF-AE architecture

To leverage the knowledge gained from an RF model, we modify the traditional AE architecture to incorporate the RF's learning. The forest-generated proximity measures [5], which indicate similarities between data points relative to the supervised task, serve as a foundation for extending the embedding while integrating the insights acquired through the RF's learning process. In RF-AE, the original input vectors $\mathbf{x}_i \in \mathbb{R}^D$ used in the vanilla AE are now replaced with the row-normalized RF-GAP proximity vector between training instance $\mathbf{x}_i$ and all the other training instances, including itself. That is, each input $\mathbf{x}_i$ used for training is now represented as an $N$-dimensional vector $\mathbf{p}_i$ encoding local-to-global supervised neighbourhood information around $\mathbf{x}_i$, defined using Eq. 2:

$$\mathbf{p}_i = [\tilde{p}(\mathbf{x}_i, \mathbf{x}_1) \quad \cdots \quad \tilde{p}(\mathbf{x}_i, \mathbf{x}_N)] \in [0, 1]^N.$$

Since its elements sum to one, $\mathbf{p}_i$ contains one-step transition probabilities from training observation with index $i$ to its supervised neighbors indexed $j = 1, \ldots, N$ derived from the RF-GAP proximities. Thus, the encoder $f(\mathbf{p}_i) = \mathbf{z}_i \in \mathbb{R}^d$ and decoder $g(\mathbf{z}_i) = \hat{\mathbf{p}}_i$ of the unconstrained RF-AE network are trained through stochastic gradient descent by minimizing the reconstruction loss $L(f, g) = \frac{1}{N} \sum_{i=1}^{N} L_{recon}(\mathbf{p}_i, \hat{\mathbf{p}}_i)$. Given a learned set of low-dimensional manifold embeddings $G = \{\mathbf{z}_i^G \in \mathbb{R}^d \mid i = 1, \ldots, N\}$ (e.g. obtained from RF-PHATE), we additionally force the RF-AE to learn a latent representation $\mathbf{z}_i$ similar to its precomputed counterpart $\mathbf{z}_i^G$ via an explicit geometric constraint to the bottleneck layer, similar to GRAE [14]. This translates into an added term in the loss formulation, which now takes the form:

$$L(f, g) = \frac{1}{N} \sum_{i=1}^{N} \left[ \lambda L_{recon}(\mathbf{p}_i, \hat{\mathbf{p}}_i) + (1 - \lambda) L_{geo}(\mathbf{z}_i, \mathbf{z}_i^G) \right].$$

The parameter $\lambda \in [0, 1]$ controls the degree to which the precomputed embedding is used in encoding $\mathbf{x}_i$: $\lambda = 1$ is our vanilla RF-AE model without geometric regularization, while $\lambda = 0$ reproduces $\mathbf{z}_i^G$ as in the standard kernel mapping formulation. We use the standard Euclidean distance for the geometric loss to align with the least-squares formulation in Eq. 1. While one could define the reconstruction loss as the squared Euclidean distance between input vectors $\mathbf{p}_i$ and their reconstructions, this biases learning toward zero-valued entries, which dominate in large datasets but carry little structural meaning. In contrast, nonzero entries reflect meaningful links in the RF-GAP graph. Although re-weighting the loss to emphasize nonzeros is possible [28], it introduces extra hyperparameters. Instead, we treat $\mathbf{p}_i$ and its reconstruction $\hat{\mathbf{p}}_i = (g \circ f)(\mathbf{p}_i)$ as probability distributions and use the Jensen-Shannon Divergence (JSD) [47] as the reconstruction loss:

$$L_{recon}(\mathbf{p}_i, \hat{\mathbf{p}}_i) = \text{JSD}(\mathbf{p}_i \,\|\, \hat{\mathbf{p}}_i), \qquad L_{geo}(\mathbf{z}_i, \mathbf{z}_i^G) = \|\mathbf{z}_i - \mathbf{z}_i^G\|_2^2.$$

In this work, we use the JSD reconstruction loss because of its ability to capture both local and global RF-GAP neighborhoods [48]. Other divergences may also be worth exploring. For example, the Kullback–Leibler (KL) divergence [49] is a natural candidate: RF-GAP already encodes some global structure through its adaptive kernel, and KL places more emphasis on accurately reconstructing nonzero inputs, which fits well with our setting. That said, the exact choice of divergence is not critical for our main contribution. The key idea is that reconstructing RF-GAP vectors and training the autoencoder to match them leads to improved structural preservation. Exploring alternative divergences is left for future work. We set the latent dimension $d = 2$ to emphasize on visual interpretability. We use RF-PHATE as the geometric constraint due to its effectiveness in supervised data visualization [4, 50], although any dimensionality reduction method can be extended this way. Refer to Fig. 1 for a comprehensive illustration of our RF-AE architecture.

### 3.3 Input dimensionality reduction with class-wise prototype selection

The input dimensionality of our RF-AE architecture scales with the training size $N$, which may cause memory issues during GPU-optimized training when dealing with large training sets. Thus we further reduce the input dimensionality of $\mathbf{p}_i$ from $N$ to $N^* \ll N$ by selecting $N^*$ prototypes. The prototypes are selected using uniform class-wise $k$-medoids [51, 52] on the induced RF-GAP training dissimilarities. First, we max-normalize the symmetrized RF-GAP proximities $p'(\mathbf{x}_i, \mathbf{x}_j) = [p(\mathbf{x}_i, \mathbf{x}_j) + p(\mathbf{x}_j, \mathbf{x}_i)]/2$ to form the symmetric dissimilarity matrix $[\max_{u,v} \{p'(\mathbf{x}_u, \mathbf{x}_v)\} - p'(\mathbf{x}_i, \mathbf{x}_j)] \in [0,1]^{N \times N}$. Then, for a dataset with $q$ classes, we find $k = N^*/q$-medoids for each class using their corresponding RF-GAP dissimilarities as input to FasterPAM [53, 54]. Let $\mathfrak{M} = \{\mathfrak{m}_1, \ldots, \mathfrak{m}_{N^*}\}$ denote the resulting set of medoid indices. Then instead of using RF-GAP transition probabilities from any point $i$ to every training point $j$ as before, we form RF-GAP transition probabilities from any point $i$ to each prototype $j \in \mathfrak{M}$ as

$$\mathbf{p}_i^* = [\tilde{p}^*(\mathbf{x}_i, \mathbf{x}_{\mathfrak{m}_1}) \quad \cdots \quad \tilde{p}^*(\mathbf{x}_i, \mathbf{x}_{\mathfrak{m}_{N^*}})], \quad \tilde{p}^*(\mathbf{x}_i, \mathbf{x}_j) = \frac{p(\mathbf{x}_i, \mathbf{x}_j)}{\sum_{j \in \mathfrak{M}} p(\mathbf{x}_i, \mathbf{x}_j)}.$$

Fig. 1 contextualizes this prototype selection mechanism within our RF-AE architecture. We also note that using prototypes allows for faster OOS projections since we no longer need to compute RF-GAP proximities to all training points.

### 3.4 Quantifying supervised OOS embedding fit

Beyond standard $k$-NN accuracy [12, 15, 34, 55, 56], which evaluates class separability in the embedding space, it is equally important to assess how well the embedding preserves the structure of informative features. Without this, class-conditional methods that artificially inflate separation may be favored, even if they distort meaningful feature–label relationships. Conversely, purely unsupervised criteria—such as neighbor preservation [12] or global distance correlation [57]—can undervalue supervised models that discard irrelevant features aligned with the classification task.

Inspired by Rhodes et al. [4], we formalize *structural importance alignment*, which quantifies the correlation between feature importances for classification and for structure preservation. Given a training/test split $X = X_{\text{train}} \cup X_{\text{test}}$ with labels $Y = Y_{\text{train}} \cup Y_{\text{test}}$, and embeddings $f_{\text{emb}}(X) = f_{\text{emb}}(X_{\text{train}}) \cup f_{\text{emb}}(X_{\text{test}}) = Z_{\text{train}} \cup Z_{\text{test}}$ from a trained encoder $f_{\text{emb}}$, we define test–train distance matrices in the original and embedded spaces as:

$$\mathbf{D}_{\text{test}}[i,j] = \|\mathbf{x}_i^{\text{test}} - \mathbf{x}_j^{\text{train}}\|_2, \quad \mathbf{D}_{\text{test}}^{\text{emb}}[i,j] = \|\mathbf{z}_i^{\text{test}} - \mathbf{z}_j^{\text{train}}\|_2, \quad \mathbf{D}_{\text{test}}, \mathbf{D}_{\text{test}}^{\text{emb}} \in \mathbb{R}_+^{N_{\text{test}} \times N_{\text{train}}}.$$

**Classification importances** are computed using a user-defined classifier $f_{\text{cls}} : \mathbb{R}^D \to \mathcal{Y}$ trained on $X_{\text{train}}$. Let $\text{acc}_{f_{\text{cls}}}(X_{\text{test}}, Y_{\text{test}})$ denote its accuracy on the test set. Then, the importance of feature $i$ is:

$$\mathcal{C}_i = \text{acc}_{f_{\text{cls}}}(X_{\text{test}}, Y_{\text{test}}) - \text{acc}_{f_{\text{cls}}}(\widetilde{X}_{\text{test}}^{(i)}, Y_{\text{test}}),$$

where $\widetilde{X}_{\text{test}}^{(i)}$ is the perturbed test set in which feature $i$ and its correlated features are permuted across samples (see Algorithm S1 in Appendix D).

**Structural importances** are computed using an unsupervised *structure preservation score* $s(\cdot, \cdot)$ that quantifies how well an embedding preserves pairwise relationships from the original space. Higher scores indicate better preservation of structure. We consider several commonly used definitions of $s$,

including local scores $s \in \{QNX, Trust\}$ and global scores $s \in \{Spear, Pearson\}$ [12, 57, 58, 59, 60]. Full definitions are provided in Appendix E.1.

Given a test set $\mathbf{D}_{\text{test}}$ and its embedding $\mathbf{D}_{\text{test}}^{\text{emb}}$, the importance of feature $i$ is then defined as:

$$\mathcal{S}_i = s(\mathbf{D}_{\text{test}}, \mathbf{D}_{\text{test}}^{\text{emb}}) - s(\widetilde{\mathbf{D}}_{\text{test}}^{(i)}, \mathbf{D}_{\text{test}}^{\text{emb}}),$$

where $\widetilde{\mathbf{D}}_{\text{test}}^{(i)}$ is the perturbed distance matrix obtained by replacing feature $i$ in $X_{\text{test}}$ with noise (Algorithm S1), while holding $X_{\text{train}}$ fixed. A larger drop in $s$ indicates that the OOS embedding relies more heavily on the structure induced by feature $i$.

To assess whether the embedding structure supports classification-relevant features, we compute the alignment between structural importances $\mathcal{S} = \{\mathcal{S}_1, \ldots, \mathcal{S}_D\}$ and classification importances $\mathcal{C} = \{\mathcal{C}_1, \ldots, \mathcal{C}_D\}$ using the Kendall rank correlation coefficient $\tau(\mathcal{C}, \mathcal{S}) \in [-1, 1]$ [61]. Higher values indicate that the embedding prioritizes features most relevant to the classification task. Fig. S2 (Appendix E.2) illustrates this Structural Importance Alignment (SIA) framework.

Note that SIA depends on both the choice of classifier $f_{\text{cls}}$ and structure score $s$. For $f_{\text{cls}}$, we use an ensemble with equal-weight majority voting across $k$-NN, SVM, and MLP classifiers to reduce model-specific bias (see Appendix G for hyperparameters). Each dataset achieves at least 60% accuracy (Appendix E.3). For $s$, we report four variants of SIA based on the chosen structure score, capturing both local and global structure preservation.

## 4 Results

### 4.1 RF-AE balances structural importance alignment and class separability

We assessed the trade-off between SIA and $k$-NN classification accuracy achieved by RF-AE against several baseline methods across 20 datasets spanning diverse domains. Each dataset contained a minimum of 1,000 samples and at least 10 features. Training and OOS embeddings were generated using an 80/20 stratified train/test split, except for Isolet, Landsat Satellite, Optical Digits, USPS, HAR, OrganC MNIST and Blood MNIST, where predefined splits were used. We applied min-max normalization to the input features prior to training and inference. Detailed descriptions of the datasets are provided in Appendix F.

Table 1 shows average SIA and $k$-NN accuracy scores across 20 datasets and 10 repetitions. We report separate local ($s = QNX, Trust$) and global ($s = Spear, Pearson$) SIA scores. Accuracy is averaged over $k = 5$ to $\sqrt{N_{\text{train}}}$ (in steps of 10) to better reflect global class separability and penalize class fragmentation. We compared RF-AE with $\lambda = 0.01$ and $N^* = 0.1 N_{\text{train}}$ to 13 baselines, including the default RF-PHATE linear kernel extension [3] (Section 2.2), vanilla AE, principal component analysis (PCA), supervised PCA, parametric $t$-SNE (P-TSNE [15, 62]), parametric UMAP (P-UMAP [12, 62]), parametric supervised UMAP (P-SUMAP [12]), pairwise controlled manifold approximation projection (PACMAP [56]), CE [34], CEBRA [63], SSNP [35] using ground-truth labels, neighborhood component analysis (NCA [55]), and partial least squares discriminant analysis (PLS-DA [64, 65]). All externally sourced methods were run using their default hyperparameter settings, as specified in the original implementations. See Appendix G.1 for full experimental details. The compute resources required for the experiments include a GPU with at least 40 GB of memory and a CPU with 128 GB of RAM, further details are provided in Appendix G.2.

Unsurprisingly, unsupervised methods generally rank lower than supervised approaches in terms of $k$-NN classification accuracy. However, even among high-accuracy models such as RF-PHATE, SSNP and P-SUMAP, we observe a notable drop in local and global SIA scores. This suggests an overemphasis on class separability at the expense of preserving the underlying supervised structure. On the other hand, unsupervised methods also struggle with SIA metrics, which is expected given their objective to preserve unsupervised pairwise similarities that may be influenced by irrelevant or noisy features. In contrast, RF-AE achieves the highest $k$-NN accuracy by a substantial margin, while consistently ranking in the top two across both local and global SIA scores. This demonstrates RF-AE's ability to not only ensure class separation but also preserve meaningful supervised relationships, effectively emphasizing the most relevant features for the classification task.

RF-AE maintains strong performance across all evaluation metrics when varying $\lambda$, as shown in Table 2. Additional ablation studies (Appendix H.1) confirm that RF-AE's performance is

Table 1: Local ($s = QNX, Trust$) and global ($s = Spear, Pearson$) SIA scores, along with test $k$-NN accuracies for our RF-AE method and 13 baselines. Scores are shown as mean $\pm$ std across 20 datasets and 10 repetitions. Methods are sorted according to accuracy. Top three values per metric are highlighted in blue, using underlined bold (first) and bold (second). In the case of ties, methods are further ranked by their standard deviations. Supervised methods are marked by an asterisk.

| | LOCAL SIA | | GLOBAL SIA | | |
| --- | --- | --- | --- | --- | --- |
| | QNX | TRUST | SPEAR | PEARSON | $k$-NN ACC |
| RF-AE* | **0.809 ± 0.024** | 0.822 ± 0.022 | **0.782 ± 0.041** | **0.779 ± 0.042** | **0.861 ± 0.009** |
| RF-PHATE* | 0.798 ± 0.025 | **0.825 ± 0.023** | 0.748 ± 0.038 | 0.750 ± 0.040 | **0.816 ± 0.010** |
| SSNP* | 0.760 ± 0.047 | 0.772 ± 0.045 | 0.685 ± 0.089 | 0.694 ± 0.080 | 0.809 ± 0.030 |
| P-SUMAP* | 0.756 ± 0.028 | 0.768 ± 0.025 | 0.647 ± 0.048 | 0.647 ± 0.048 | 0.797 ± 0.011 |
| CE* | 0.795 ± 0.050 | 0.818 ± 0.051 | 0.765 ± 0.051 | **0.763 ± 0.054** | 0.797 ± 0.043 |
| NCA* | **0.808 ± 0.027** | 0.805 ± 0.025 | **0.771 ± 0.032** | 0.759 ± 0.033 | 0.760 ± 0.007 |
| PACMAP | 0.749 ± 0.026 | 0.758 ± 0.025 | 0.688 ± 0.029 | 0.688 ± 0.029 | 0.743 ± 0.011 |
| P-TSNE | 0.743 ± 0.028 | 0.747 ± 0.028 | 0.684 ± 0.036 | 0.666 ± 0.038 | 0.712 ± 0.012 |
| AE | 0.744 ± 0.027 | 0.751 ± 0.029 | 0.695 ± 0.044 | 0.655 ± 0.053 | 0.700 ± 0.018 |
| P-UMAP | 0.757 ± 0.027 | 0.744 ± 0.028 | 0.674 ± 0.035 | 0.657 ± 0.038 | 0.655 ± 0.022 |
| SPCA* | 0.767 ± 0.026 | 0.759 ± 0.030 | 0.741 ± 0.031 | 0.738 ± 0.032 | 0.624 ± 0.009 |
| PLS-DA* | 0.715 ± 0.026 | 0.708 ± 0.028 | 0.659 ± 0.027 | 0.639 ± 0.028 | 0.592 ± 0.009 |
| CEBRA* | 0.780 ± 0.045 | 0.775 ± 0.050 | 0.735 ± 0.062 | 0.728 ± 0.068 | 0.582 ± 0.040 |
| PCA | 0.745 ± 0.027 | 0.742 ± 0.026 | 0.733 ± 0.027 | 0.727 ± 0.028 | 0.563 ± 0.009 |

robust to both $\lambda$ and prototype count $N^*$. However, ablation experiments in Appendix H.2 with alternative geometric regularizers show that these methods consistently underperform compared to our proposed RF-PHATE. Still, RF-based variants such as RF-UMAP better align with the RF-GAP reconstruction loss during training, making them interesting candidates for further exploration. Finally, RF-AE's superiority in SIA persists under varying classification importance strategies (Appendix H.3), reflecting stronger alignment with the ground-truth feature importance hierarchy.

Table 2: Comparison of SIA scores and $k$-NN accuracy across different $\lambda$ values for RF-AE. Each score is compared with baseline models in Table 1, and highlighted only if it ranks among the top three overall. Top three values per metric are highlighted in blue, using underlined bold (first) and bold (second). RF-AE demonstrates strong robustness across a broad range of $\lambda$ values.

| | LOCAL SIA | | GLOBAL SIA | | |
| --- | --- | --- | --- | --- | --- |
| $\lambda$ | QNX | TRUST | SPEAR | PEARSON | $k$-NN ACC |
| 0 | **0.812 ± 0.025** | 0.822 ± 0.023 | 0.751 ± 0.038 | 0.752 ± 0.040 | **0.836 ± 0.011** |
| 0.001 | **0.809 ± 0.024** | **0.824 ± 0.024** | 0.763 ± 0.037 | 0.757 ± 0.039 | **0.859 ± 0.009** |
| 0.01 | **0.809 ± 0.024** | 0.822 ± 0.022 | **0.782 ± 0.041** | **0.779 ± 0.042** | **0.861 ± 0.009** |
| 0.1 | **0.808 ± 0.023** | 0.822 ± 0.021 | **0.782 ± 0.060** | **0.784 ± 0.054** | **0.864 ± 0.009** |
| 1 | **0.804 ± 0.023** | 0.820 ± 0.022 | 0.689 ± 0.102 | 0.686 ± 0.108 | **0.864 ± 0.009** |

## 4.2 Qualitative comparison through OOS visualizations

We qualitatively assessed the capability of four methods to embed OOS instances on Sign MNIST (A–K) and OrganC MNIST dataset. Each model was trained on the training subset, and we subsequently mapped the test set with the learned encoder. Fig. 2 depicts the resulting visualizations.

From the Sign MNIST (A–K) plot in Fig. 2a, RF-AE retains the overall shape of the RF-PHATE embedding while providing finer within-class detail. In contrast, RF-PHATE's default kernel extension compresses class clusters, making local relationships harder to discern. RF-AE expands these clusters, revealing within-class patterns, such as the logical transition between variations in shadowing and hand orientation to represent the letter "C" (top-right cluster). This detail is lost in RF-PHATE's default kernel extension, which over-relies on diffusion and smooths out subtle differences. P-TSNE captures local structure but fragments same-class samples based on irrelevant

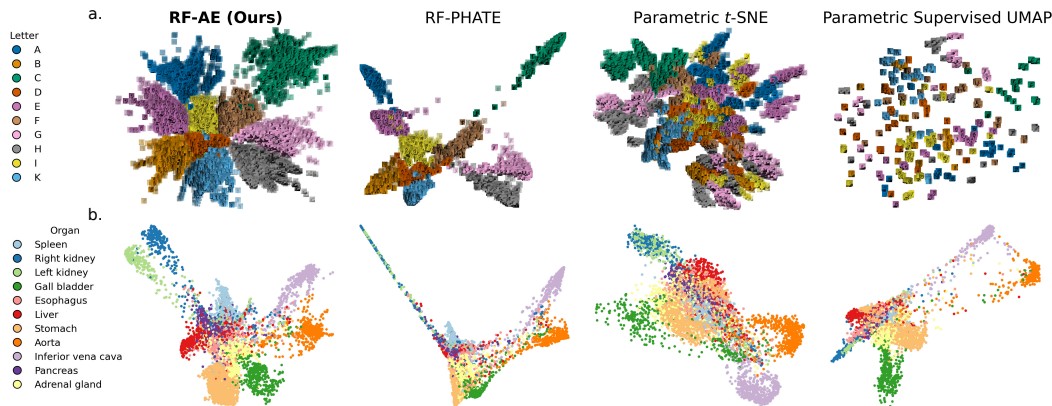

Figure 2: OOS visualization using four different dimensionality reduction methods. **a.** Sign MNIST (A–K) dataset (Table S2): Training and test samples are shown with their original images, color-tinted by label. Training samples appear with reduced opacity. RF-AE captures supervised relationships by preserving class-specific variations, such as shadowing and hand orientation, while also highlighting inter-class similarities and maintaining clear class separability. The default RF-PHATE's kernel extension compresses clusters excessively. Parametric $t$-SNE and parametric supervised UMAP demonstrate sensitivity to irrelevant features. **b.** OrganC MNIST dataset: Test points are shown as color-coded circles based on their labels. Training points are omitted for clarity. RF-AE clearly separates similar organ types while preserving their anatomical proximity, capturing both class identity and biological relevance. Other methods tend to merge these classes or distort their relationships, failing to reflect fine-grained anatomical distinctions.

background variations—for example, grouping shadowed "G" and "H" instances together (bottom cluster) while separating them from unshadowed counterparts (far-left cluster). RF-AE avoids this issue, distinguishing within-class variations while preserving the class-specific clusters. P-SUMAP is also sensitive to irrelevant features and tends to artificially cluster neighboring points of the same class, leading to a sparse and fragmented embedding.

From the OrganC MNIST plot in Fig. 2b, RF-AE yields the most structured and interpretable embedding, forming well-separated clusters while preserving anatomical proximity—e.g., between the left and right kidneys, or among the stomach, liver, and pancreas. RF-PHATE captures smooth transitions but merges nearby classes like the left and right kidneys, reducing local separability. P-TSNE shows significant class overlap due to the absence of supervision, with only four out of eleven classes clearly separated. While P-SUMAP offers better class separability than P-TSNE, its projections remain difficult to interpret, with elongated structures (e.g., aorta and inferior vena cava) and compact, overlapping anatomical regions near the center that blur class boundaries.

This qualitative analysis of the Sign MNIST and OrganC MNIST dataset underscores the importance of regularization and methodological choices in creating meaningful embeddings for supervised tasks. RF-AE, guided by RF-PHATE regularization, effectively preserves both local and global structure, outperforming existing methods in capturing informative patterns. Additional visualizations on audio data and extended comparisons with the other models are provided in Appendices I.1 and I.2.

## 5 Discussion

The significance of supervised dimensionality reduction lies in its ability to reveal meaningful relationships between features and labels. As shown by Rhodes et al. [4], RF-PHATE stands out as a strong solution for supervised data visualization. However, it lacks an embedding function for OOS extension. To address this limitation, we designed Random Forest Autoencoders (RF-AE), an autoencoder-based architecture that reconstructs RF-GAP neighborhoods while preserving the supervised geometry captured by precomputed RF-PHATE embeddings. Our experiments confirmed the utility of this extension, demonstrating its ability to embed new data points while retaining the intrinsic supervised manifold structure. We quantitatively showed that RF-AE with RF-PHATE regularization outperforms baseline kernel extensions and other parametric embedding models in

generating OOS embeddings that preserve feature importances relevant to the classification task while maintaining class separability. We further showed in Appendix H.1 that RF-AE's performance is robust to the geometric constraint $\lambda$ and the number of selected prototypes $N^*$. Visually, we observed that RF-AE regularized by RF-PHATE inherits the denoised local-to-global supervised structure of RF-PHATE while increasing resolution for improved within-class visualization. This results in a more effective tradeoff between preserving informative structure and achieving class separability than baseline RF-PHATE kernel extensions. Other methods either over-emphasize class separability or fail to incorporate sufficient supervision, leading to noisier visualizations. Finally, RF-AE can project unseen data without labels, support semi-supervised training (Appendix J), and handle any data modality without additional preprocessing, making it well suited for semi-supervised tasks across diverse applications.

While effective, RF-AE inherits scalability constraints from the computational cost of the full RF-GAP proximity matrix during training (Appendix K). Our prototype selection strategy already yields substantial speedups over RF-PHATE's default kernel extension at inference, while upcoming vectorized and sparse RF-GAP computation (to be released soon on GitHub) will further reduce this bottleneck. Future work includes investigating RF-AE's efficiency in semi-supervised hierarchical clustering and adaptive methods for prototype selection [52]. Building on our ablation studies (Appendix H.2), researchers may also explore other RF-based geometric constraints.

## Broader impacts

This paper advances guided data representation learning by integrating expert-derived annotations and enabling out-of-sample extension, thus allowing generalization beyond the training set. Nonetheless, we advise users to interpret supervised 2D visualizations with caution, as label assignments may introduce biases. When labels reflect social or demographic factors, supervised methods are prone to embedding structural biases since they explicitly aim to discriminate between classes. Bias can also arise in highly imbalanced settings: the underlying Random Forest tends to favor majority classes, which can cause minority classes to appear artificially closer to or farther from other groups, as the features that characterize them may not be adequately captured in the RF-GAP proximities. These concerns are not unique to RF-AE but extend to supervised methods in general. That said, RF-AE helps mitigate such issues by avoiding the exaggerated separations often produced by purely class-conditional approaches, making it a comparatively more reliable choice in practice.

Acknowledging these limitations, our method still offers valuable support to decision-makers by providing interpretable visualizations while remaining scalable and applicable in (semi-)supervised tasks. In particular, RF-AE can assist expert- or AI-based disease diagnosis by projecting incoming patient instances into a 2D space, where they can be contextualized relative to existing embeddings. Such visualizations allow practitioners to assess whether a prediction is consistent with established structures or deviates from them, offering a practical indicator of prediction reliability. Overall, RF-AE has potential societal impact in biomedical research, as well as broader applications for data-driven insights in healthcare, finance, and multimedia.

## Acknowledgments and Disclosure of Funding

We thank the anonymous reviewers for their meaningful feedback, which led us to improve our paper. This research was enabled in part by compute resources provided by Mila – Quebec AI Institute (mila.quebec) and the Department of Mathematics and Statistics (Université de Montréal). This research was supported in part by the Ministry of Health and Social Services (Quebec) in collaboration with the Centre intégré de santé et de services sociaux Centre-Sud-de-l'Île-de-Montréal [A.A.], Canada CIFAR AI Chair [G.W.], NSERC Discovery grant 03267 [G.W.], NIH grant R01GM135929 [G.W.], NSF grant DMS-2327211 [G.W.], NSF grant 221232 [K.M], FRQNT Doctoral research scholarship [M.L.] and the IVADO Visiting Scholar program [K.M]. The authors are solely responsible for the content of this work. The views expressed do not necessarily reflect those of the funding agencies. The funders had no involvement in the study design, data collection and analysis, decision to publish, or manuscript preparation. The authors declare no competing interests.

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

# A  Maximality of the RF-GAP self-similarity

Recall the RF-GAP proximity (Section 3.1) between observations $\mathbf{x}_i$ and $\mathbf{x}_j$

$$
p(\mathbf{x}_i, \mathbf{x}_j) = \begin{cases} \dfrac{1}{|\bar{S}_i|} \sum_{t \in \bar{S}_i} \dfrac{c_i(t)}{|M_i(t)|}, & j = i, \\[3ex] \dfrac{1}{|S_i|} \sum_{t \in S_i} \dfrac{c_j(t)\, I\,(j \in J_i(t))}{|M_i(t)|}, & j \neq i, \end{cases}
$$

where:

- $S_i$ is the set of trees in which $\mathbf{x}_i$ is *out-of-bag* (OOB).
- $\bar{S}_i$ is the set of trees in which $\mathbf{x}_i$ is *in-bag*.
- $c_i(t)$ is the bootstrap multiplicity of $\mathbf{x}_i$ in tree $t$.
- $J_i(t)$ is the set of in-bag points that share the terminal node with $\mathbf{x}_i$ in tree $t$.
- $M_i(t)$ is the multiset of in-bag sample indices in that terminal node (counting multiplicities).

For any fixed tree $t$, we define the random quantities

- $B_i(t) = I(\mathbf{x}_i$ is in-bag in tree $t)$,
- $D_{ij}(t) = I(B_j(t) = 1$ and $\mathbf{x}_j$ shares $\mathbf{x}_i$'s leaf in tree $t)$,
- $c_i(t) \sim \text{Binomial}\left(N, \frac{1}{N}\right), \quad i = 1, \dots, N,$

Thus, considering all trees in the forest,

- $T_i^{\text{IB}} = \sum_{t=1}^{|T|} B_i(t)$,
- $T_i^{\text{OOB}} = \sum_{t=1}^{|T|} 1 - B_i(t)$,

and per-tree contributions to self- and cross-similarity are re-written as

$$
\alpha_{ii}(t) := B_i(t)\, \frac{c_i(t)}{|M_i(t)|}, \qquad \alpha_{ij}(t) := D_{ij}(t)\, \frac{c_j(t)}{|M_i(t)|} \quad (i \neq j).
$$

Under the standard Random Forests assumptions, the following holds:

- *Tree independence.* Each tree is grown from an independent bootstrap sample and an independent sequence of feature splits, ensuring i.i.d. per-tree contributions $\alpha_{ii}(t)$ and $\alpha_{ij}(t)$.
- *Bootstrap inclusion probability.* An observation is in-bag in tree $t$ with probability

$$
p := \mathbb{P}[B_i(t) = 1] = \mathbb{P}[c_i(t) \geq 1] = 1 - \mathbb{P}[c_i(t) = 0] = 1 - \left(1 - \tfrac{1}{N}\right)^N
$$
$$
\longrightarrow 1 - e^{-1} \approx 0.632.
$$

  Hence,
$$
\begin{aligned} B_i(t) &\sim \text{Bernoulli}(p) \\ T_i^{\text{IB}} &\sim \text{Binomial}(|T|, p) \\ T_i^{\text{OOB}} &\sim \text{Binomial}(|T|, 1-p) \end{aligned}
$$

- *Co-occurrence probability.* Even if $\mathbf{x}_j$ is very similar to $\mathbf{x}_i$, the probability that they end up together in the same leaf *and* $\mathbf{x}_i$ was OOB is strictly less than 1:

$$
q_{ij} := \mathbb{P}\left[D_{ij}(t) = 1 \mid B_i(t) = 0\right] < 1 \quad (i \neq j).
$$

**Proposition A.1.** *For every fixed $i$ and any $j \neq i$, in the limit as the number of trees $|T| \to \infty$,*

$$
p(\mathbf{x}_i, \mathbf{x}_i) > p(\mathbf{x}_i, \mathbf{x}_j)
$$

*Proof.* RF-GAP similarities are re-written as random variables:

$$p(\mathbf{x}_i, \mathbf{x}_j) = \begin{cases} \dfrac{1}{T_i^{\mathrm{IB}}} \displaystyle\sum_{t=1}^{T} \alpha_{ii}(t), & \text{if } i = j, \\[2ex] \dfrac{1}{T_i^{\mathrm{OOB}}} \displaystyle\sum_{t=1}^{T} \alpha_{ij}(t), & \text{otherwise.} \end{cases}$$

By the Strong Law of Large Numbers and tree-independence, as $|T| \to \infty$ we have almost surely

$$\frac{1}{|T|} \sum_{t=1}^{|T|} \alpha_{ii}(t) \longrightarrow \mathbb{E}[\alpha_{ii}(t)] = \mathbb{E}\left[ B_i(t) \frac{c_i(t)}{|M_i(t)|} \right] = p \underbrace{\mathbb{E}\left[ \frac{c_i(t)}{|M_i(t)|} \mid B_i(t) = 1 \right]}_{=\mu} = p\,\mu,$$

$$\begin{aligned} \frac{1}{|T|} \sum_{t=1}^{|T|} \alpha_{ij}(t) \longrightarrow \mathbb{E}[\alpha_{ij}(t)] &= \mathbb{E}\left[ D_{ij}(t) \frac{c_j(t)}{|M_i(t)|} \right] \\ &= (1-p)\, q_{ij}\, \underbrace{\mathbb{E}\left[ \frac{c_j(t)}{|M_i(t)|} \mid D_{ij}(t) = 1, B_i(t) = 0 \right]}_{\leq \mu} \\ &\leq (1-p)\, q_{ij}\, \mu. \end{aligned}$$

The inequality $\mathbb{E}\left[ \frac{c_j(t)}{|M_i(t)|} \mid D_{ij}(t) = 1, B_i(t) = 0 \right] \leq \mu := \mathbb{E}\left[ \frac{c_i(t)}{|M_i(t)|} \mid B_i(t) = 1 \right]$ follows from the fact that while $c_j(t)$ and $c_i(t)$ have identical marginal distributions, the conditional distribution of the shared leaf size $|M_i(t)|$ is stochastically larger under the event $D_{ij}(t) = 1$, $B_i(t) = 0$ than under $B_i(t) = 1$ alone. Indeed, conditioning on $D_{ij}(t) = 1$ requires that the in-bag point $\mathbf{x}_j$ and the out-of-bag point $\mathbf{x}_i$ fall in the same leaf, which favors larger leaves with broader decision rules. This increases the expected denominator $|M_i(t)|$ and thereby reduces the expected normalized multiplicity $c_j(t)/|M_i(t)|$. Moreover, almost surely,

$$\frac{T_i^{\mathrm{IB}}}{|T|} \longrightarrow \mathbb{E}\left[ B_i(t) \right] = p, \qquad \frac{T_i^{\mathrm{OOB}}}{|T|} \longrightarrow \mathbb{E}\left[ 1 - B_i(t) \right] = 1 - p$$

Thus,

$$p(\mathbf{x}_i, \mathbf{x}_i) = \frac{\frac{1}{|T|} \sum_{t=1}^{|T|} \alpha_{ii}(t)}{\frac{T_i^{\mathrm{IB}}}{|T|}} \longrightarrow \frac{p\,\mu}{p} = \mu,$$

$$p(\mathbf{x}_i, \mathbf{x}_j) = \frac{\frac{1}{|T|} \sum_{t=1}^{|T|} \alpha_{ij}(t)}{\frac{T_i^{\mathrm{OOB}}}{|T|}} \longrightarrow \frac{\mathbb{E}[\alpha_{ij}(t)]}{1 - p} \leq \frac{(1-p)\, q_{ij}\, \mu}{1 - p} = q_{ij}\, \mu.$$

Since we assumed $q_{ij} < 1$, it follows that

$$\mu > q_{ij}\, \mu \implies \lim_{|T| \to \infty} p(\mathbf{x}_i, \mathbf{x}_i) > \lim_{|T| \to \infty} p(\mathbf{x}_i, \mathbf{x}_j).$$

$\square$

*Remark* A.2. Finite-$|T|$ concentration bounds (e.g. Hoeffding's inequality [66]) imply the same inequality holds with overwhelming probability.

*Remark* A.3. The assumption $q_{ij} < 1$ is not necessary for non-strict inequality.

## B   RF-GAP representations stabilize supervised manifold learning

We designed our RF-AE framework under the premise that encoders operating on (supervised) kernel representations are better suited for supervised settings than those using raw input features. This

assumption stems from the ability of well-chosen kernel functions to effectively filter out irrelevant features, thereby enhancing the encoder's robustness to highly noisy datasets. To empirically validate this, we conducted a toy experiment using the artificial tree dataset described in Appendix C. To simulate a noisy input space, we progressively augmented the dataset with additional features sampled from a uniform distribution $U(0, 1)$, corresponding to various signal-to-noise ratios (SNR) $\in \{\infty, 1, 0.1, 0.01, 0.001\}$. We then randomly selected 80% of each augmented dataset to train both models to regress onto the precomputed training RF-PHATE embeddings. The two MLP regressors shared the exact same architecture and hyperparameters (Appendix G), differing only in their input representations. We evaluated the trained models on the remaining 20% test split and visualized their two-dimensional embeddings under different SNR conditions, along with the ground-truth tree structure and median learning curves over 50 epochs across 10 repetitions, as shown in Fig. S1.

The training RF-PHATE embeddings accurately capture the underlying ground-truth structure, making them a strong supervisory signal for manifold learning. Our RF-GAP-based encoder proves highly robust to irrelevant features, producing well-structured embeddings even under severe noise conditions (e.g., SNR = 0.001). It consistently converges faster and reaches a better local minimum without overfitting, as evidenced by its test embeddings (middle row), which closely mirror the ground-truth structure. In contrast, the feature-based MLP is much more sensitive to noise, with increasing training loss and disordered embeddings in both training and test sets. Even under low-noise settings (SNR $= \infty$ or 1), it fails to achieve comparable performance, highlighting the superior robustness and generalization ability of our kernel-based encoder.

## C   Artificial tree construction

We constructed the artificial tree data used in Appendix B following the method described in the original PHATE paper [3]. The first branch of the tree consists of 100 linearly spaced points spanning four dimensions, with all other dimensions set to zero. The second branch starts at the endpoint of the first branch, with its 100 points remaining constant in the first four dimensions while progressing linearly in the next four dimensions, leaving all others at zero. Similarly, the third branch progresses linearly in dimensions 9–12, with subsequent branches following the same pattern but differing in length, resulting in 40 dimensions. Each branch endpoint and branching point includes an additional 40 points, and zero-mean Gaussian noise (standard deviation 7) is added to simulate gene expression advancement along the branches. Before visualization, all features are normalized to the range [0, 1].

## D   Feature correlation-aware data perturbation

In this section, we detail the procedure for generating perturbed datasets using a correlation-aware random sampling strategy [67]. Since ground-truth feature importance are rarely available, this approach is employed to generate pseudo-ground-truth feature importances as part of our evaluation scheme (Section 3.4). For each feature $i$, instead of permuting feature $i$'s column values—as in the standard permutation approach—, we reassign them by randomly sampling values from the feature space. Additionally, all other feature column values are randomly replaced with a probability proportional to their absolute correlation with $i$. In other words, column values for features highly correlated with $i$ are also replaced by random sampling, while column values for features not correlated with $i$ remain unchanged. This prevents the determination of fallacious feature importances where all correlated features are assigned zero importance. Refer to Algorithm S1 for a step-by-step description of this feature-wise data perturbation procedure.

## E   Evaluation metric details for supervised OOS embedding

### E.1   Structure preservation scores

Let $\mathbf{D}_{\text{test}}, \mathbf{D}_{\text{test}}^{\text{emb}} \in \mathbb{R}_+^{N_{\text{test}} \times N_{\text{train}}}$ denote the test–train distance matrices in the original and embedded spaces, respectively. We define four structure preservation metrics $s(\mathbf{D}_{\text{test}}, \mathbf{D}_{\text{test}}^{\text{emb}})$, which are used to compute multi-view structural alignment scores introduced in Section 3.4. We categorize these metrics into local and global types and cite the reference works where they were previously used to assess the quality of embedding methods.

---

**Algorithm S1:** Feature-wise data perturbation with random sampling

---

**Input:** Input data $X$, feature correlation matrix $\mathbf{C}$
**Output:** Perturbed datasets $\tilde{\mathbf{X}}$ for each feature

**1** Initialize list $\tilde{\mathbf{X}}$ to store perturbed datasets;

**2** Generate $\tilde{X}$ from $X$ by randomly sampling column values without replacement;

**3** **foreach** *feature $i$* **do**

**4**     Generate mask matrix $\mathbf{M}$ with elements $\mathbf{M}[i,j] \in \{0,1\}$ sampled from Bernoulli$(|\mathbf{C}[i,j]|)$;

**5**     Build perturbed dataset: $\tilde{X}^i = \mathbf{M} \odot \tilde{X} + (I - \mathbf{M}) \odot X$;

**6**     Store $\tilde{\mathbf{X}}[i] = \tilde{X}^i$;

**7** **return** $\tilde{\mathbf{X}}$

---

**Local Structure Preservation Scores**

- **QNX (Quality of Neighborhood eXtrapolation)** [12, 58]:

$$QNX(\mathbf{D}_{\text{test}}, \mathbf{D}_{\text{test}}^{\text{emb}}) := \frac{1}{N_{\text{test}}} \sum_{i=1}^{N_{\text{test}}} \frac{1}{K} \sum_{j \in \mathcal{N}_K^{\text{emb}}(i)} I\left(j \in \mathcal{N}_K^{\text{true}}(i)\right),$$

where $\mathcal{N}_K^{\text{true}}(i)$ are the indices of the $K$ smallest entries in row $\mathbf{D}_{\text{test}}[i,:]$, and $\mathcal{N}_K^{\text{emb}}(i)$ are those in $\mathbf{D}_{\text{test}}^{\text{emb}}[i,:]$.

- **Trustworthiness** [59]:

$$Trust(\mathbf{D}_{\text{test}}, \mathbf{D}_{\text{test}}^{\text{emb}}) := 1 - \frac{2}{N_{\text{test}}K(2N_{\text{train}} - 3K - 1)} \sum_{i=1}^{N_{\text{test}}} \sum_{j \in \mathcal{U}_i} \left(r_{ij}^{\text{true}} - K\right),$$

where $\mathcal{U}_i = \mathcal{N}_K^{\text{emb}}(i) \setminus \mathcal{N}_K^{\text{true}}(i)$, and $r_{ij}^{\text{true}}$ is the rank of index $j$ in row $\mathbf{D}_{\text{test}}[i,:]$.

**Global Structure Preservation Scores**

- **Spearman rank correlation** [57]:

$$Spear(\mathbf{D}_{\text{test}}, \mathbf{D}_{\text{test}}^{\text{emb}}) := \text{corr}_{\text{rank}}\left(\text{vec}(\mathbf{D}_{\text{test}}),\ \text{vec}(\mathbf{D}_{\text{test}}^{\text{emb}})\right),$$

where $\text{vec}(\cdot)$ denotes vectorization and $\text{corr}_{\text{rank}}$ is the Spearman rank correlation [68].

- **Pearson correlation** [60]:

$$Pearson(\mathbf{D}_{\text{test}}, \mathbf{D}_{\text{test}}^{\text{emb}}) := \text{corr}\left(\text{vec}(\mathbf{D}_{\text{test}}),\ \text{vec}(\mathbf{D}_{\text{test}}^{\text{emb}})\right),$$

using the Pearson linear correlation [69] between flattened test–train distance vectors.

For robustness, we averaged local metrics over different neighborhood sizes, ranging from $K = 5$ to $K = \sqrt{N_{\text{train}}}$, in steps of 10.

### E.2  Illustration of our structural importance alignment framework

Fig. S2 illustrates our SIA framework for evaluating supervised OOS embedding quality using the Sign MNIST (A–K) dataset (Table S2). While both RF-AE and P-TSNE produce locally plausible embeddings, their ability to preserve class-relevant structure differs significantly. RF-AE emphasizes informative regions—such as hand and finger contours—while mitigating background effects. In contrast, P-TSNE attributes higher structural importance to background pixels, leading to poorer alignment with classification-relevant features. This discrepancy is reflected in the final local SIA scores: RF-AE achieves a much higher alignment (0.85) than P-TSNE (0.55), confirming that RF-AE better preserves the semantic structure needed for accurate classification in OOS settings. These findings support our qualitative observations from Section 4.2.

### E.3 Baseline classifiers' hyperparameters and performance

Since ground-truth classification importances are rarely available, our SIA framework (Section 3.4) uses a baseline classifier $f_{\text{cls}}$ to derive pseudo-ground-truth importances. To ensure these importances are meaningful, $f_{\text{cls}}$ must achieve reasonably high test accuracy. Table S1 reports per-dataset accuracies for both $f_{\text{cls}} = k$-NN and an ensemble classifier $f_{\text{cls}} = k$-NN + SVM + MLP combining $k$-NN, SVM, and MLP predictions via equal-weight voting. We use $k = \sqrt{N_{\text{train}}}$ for the $k$-NN classifier. The SVM is implemented using scikit-learn's LinearSVC [70] with default hyperparameters. The MLP is a two-layer feedforward network with hidden dimensions $h_1 = \lfloor \frac{2}{3} \cdot D \rfloor$ and $h_2 = \lfloor \frac{1}{3} \cdot D \rfloor$, where $D$ is the input dimensionality. Each hidden layer is followed by ReLU activation, dropout (rate 0.2), and layer normalization. Weight normalization is applied to the first two linear layers. The final layer is a standard linear projection without activation.

We find that the ensemble consistently improves upon standalone $k$-NN and achieves above 60% accuracy on all datasets, making it a suitable proxy for generating classification importances. Nonetheless, $k$-NN alone performs reasonably well, falling below 60% accuracy on OrganC MNIST dataset. For a detailed comparison of SIA scores using $k$-NN instead of the ensemble, see Section H.3.

Table S1: Average test classification accuracy (mean $\pm$ std) per dataset (see Appendix F), using a single $k$-NN classifier (left column) and an ensemble of $k$-NN, linear SVM, and MLP classifiers (right column). The ensemble generally outperforms the standalone $k$-NN, making it a robust reference for generating classification feature importances.

| DATASET | $k$-NN | $k$-NN + SVM + MLP |
|---|---|---|
| QSAR BIODEGRADATION | $0.835 \pm 0.032$ | $0.854 \pm 0.028$ |
| BLOOD MNIST | $0.742 \pm 0.000$ | $0.761 \pm 0.049$ |
| CARDIOTOCOGRAPHY | $0.662 \pm 0.026$ | $0.681 \pm 0.023$ |
| CHESS | $0.914 \pm 0.012$ | $0.942 \pm 0.012$ |
| DIABETIC RETINOPATHY DEBRECEN | $0.657 \pm 0.043$ | $0.686 \pm 0.031$ |
| FASHION MNIST (TEST) | $0.777 \pm 0.007$ | $0.827 \pm 0.007$ |
| GTZAN (3-SEC) | $0.708 \pm 0.006$ | $0.678 \pm 0.013$ |
| HAR (USING SMARTPHONES) | $0.887 \pm 0.000$ | $0.917 \pm 0.009$ |
| ISOLET | $0.906 \pm 0.000$ | $0.931 \pm 0.004$ |
| LANDSAT SATELLITE | $0.858 \pm 0.000$ | $0.829 \pm 0.010$ |
| MNIST (TEST) | $0.894 \pm 0.008$ | $0.922 \pm 0.006$ |
| OBESITY | $0.625 \pm 0.018$ | $0.661 \pm 0.026$ |
| OPTICAL BURST SWITCHING NETWORK | $0.745 \pm 0.028$ | $0.743 \pm 0.023$ |
| OPTICAL DIGITS | $0.953 \pm 0.000$ | $0.948 \pm 0.004$ |
| ORGANC MNIST | $0.473 \pm 0.000$ | $0.627 \pm 0.004$ |
| SIGN MNIST (A–K) | $0.908 \pm 0.007$ | $0.940 \pm 0.008$ |
| SPAMBASE | $0.860 \pm 0.008$ | $0.875 \pm 0.008$ |
| SPORTS ARTICLES | $0.809 \pm 0.019$ | $0.818 \pm 0.019$ |
| USPS | $0.871 \pm 0.000$ | $0.891 \pm 0.002$ |
| WAVEFORM | $0.848 \pm 0.012$ | $0.860 \pm 0.014$ |

## F   Description of the datasets

Table S2 provides additional details on the datasets used for the quantitative and qualitative comparisons between RF-AE and other methods. Sign MNIST (A–K) [71], MNIST (test subset) [72], Fashion MNIST (test subset) [73], GTZAN (3-second version) [74] and USPS [75] were obtained from Kaggle. The Sign MNIST (A–K) dataset is a subset of the original, containing the first 10 letters (excluding J, which requires motion). Blood MNIST and OrganC MNIST (Med MNIST family [76, 77]) were obtained from Zenodo. All other datasets are publicly available from the UCI Machine Learning Repository.

Table S2: Description of the 20 datasets used in our experiments, grouped by data modality.

| Dataset | Size | Test % | Dimensions | Classes |
|---|---|---|---|---|
| TABULAR / CLINICAL | | | | |
| CARDIOTOCOGRAPHY | 2126 | 0.20 | 21 | 10 |
| DIABETIC RETINOPATHY DEBRECEN | 1151 | 0.20 | 19 | 2 |
| OBESITY | 2111 | 0.20 | 16 | 7 |
| QSAR BIODEGRADATION | 1055 | 0.20 | 41 | 2 |
| TEXT / NLP | | | | |
| SPAMBASE | 4601 | 0.20 | 57 | 2 |
| SPORTS ARTICLES | 1000 | 0.20 | 59 | 2 |
| SENSOR / TIME SERIES | | | | |
| HAR (USING SMARTPHONES) | 10299 | 0.29 | 561 | 6 |
| ISOLET | 7797 | 0.20 | 617 | 26 |
| WAVEFORM | 5000 | 0.20 | 40 | 3 |
| LANDSAT SATELLITE | 6435 | 0.31 | 36 | 6 |
| IMAGE (GENERAL) | | | | |
| OPTICAL DIGITS | 5620 | 0.32 | 64 | 10 |
| USPS | 9298 | 0.22 | 256 | 10 |
| MNIST (TEST) | 10000 | 0.20 | 784 | 10 |
| FASHION MNIST (TEST) | 10000 | 0.20 | 784 | 10 |
| SIGN MNIST (A–K) | 14482 | 0.20 | 784 | 10 |
| IMAGE (BIOMEDICAL) | | | | |
| BLOOD MNIST | 15380 | 0.22 | 2352 | 8 |
| ORGANC MNIST | 21191 | 0.39 | 784 | 11 |
| AUDIO | | | | |
| GTZAN (3-SEC) | 9990 | 0.20 | 57 | 10 |
| NETWORK / TRAFFIC | | | | |
| OPTICAL BURST SWITCHING NETWORK | 1060 | 0.20 | 21 | 4 |
| GAMES / LOGIC | | | | |
| CHESS | 3196 | 0.20 | 36 | 2 |

# G   Experimental setting

## G.1   Model implementations and hyperparameters

Unless otherwise specified, all methods were run with their default hyperparameters in our experiments.

- **RF-AE:** Implemented in PyTorch. The encoder $f$ consisted of three hidden layers with sizes 800, 400, and 100. The bottleneck layer was set to dimension 2 for visualization. The decoder $g$ was symmetric with layers of sizes 100, 400, and 800, followed by an output layer matching the input dimensionality. ELU activations were used throughout, except for the bottleneck (identity) and output (softmax) layers. Training was performed using the AdamW optimizer [78] with a learning rate of $10^{-3}$, batch size of 512, weight decay of $10^{-5}$, and 200 epochs without early stopping. We set the default $\lambda$ and $N^*$ to 0.01 and $0.1 N_{\text{train}}$, respectively.

- **SSNP, CE, and vanilla AE:** Implemented using the same architecture and activations as RF-AE. For SSNP, we followed the authors' recommendations: a sigmoid output activation and a reconstruction-classification loss balance of 0.5. For CE and vanilla AE, the output activation was the identity function.

- **Parametric $t$-SNE and UMAP:** Implemented following Damrich et al. [62], using the InfoNCE loss [79]; available at https://github.com/sdamrich/cl-tsne-umap.

- **Parametric supervised UMAP:** Official implementation from https://github.com/lmcinnes/umap.

- **PaCMAP:** From `https://github.com/YingfanWang/PaCMAP`.
- **CEBRA:** From `https://github.com/AdaptiveMotorControlLab/CEBRA`. We used 200 training epochs and a batch size of 512, as recommended by the authors.
- **SPCA:** From `https://github.com/bghojogh/Principal-Component-Analysis`.
- **PCA, NCA, and PLS-DA:** Implemented using the `scikit-learn` library [70].

### G.2 Compute resources

Experiments were conducted on a shared computing environment with access to both GPU and CPU resources. For models requiring GPU acceleration, we used:

- 1 GPU with at least 40 GB of memory (e.g., NVIDIA A100 40GB, H100 80GB, or equivalent),
- 1 CPU with 128 GB of RAM.

For models that do not require GPU acceleration, computations were performed using CPU only, with a minimum of 128 GB of RAM.

We conducted experiments across 20 datasets for our RF-AE model and 13 baseline methods, using multiple random seeds to report the mean and standard deviation of performance metrics. All hyperparameters and configurations were managed using Hydra [80]. Code and configuration files will be released to ensure full reproducibility.

The runtime for RF-AE training and the entire evaluation process for individual experiments, where each experiment is defined as running one model on one dataset with a single random seed, ranged from 1 to 6 hours depending on the dataset size.

## H  Ablation experiments

### H.1  Impact of the reconstruction weight and prototype count

We performed ablation experiments on the two main RF-AE hyperparameters: the loss balancing parameter $\lambda$ and the number of selected prototypes $N^*$. We report local/global SIA scores and $k$-NN accuracies across combinations $(\lambda, N^*) \in \{1, 0.1, 0.01, 0.001, 0\} \times \{pN_{\text{train}} \mid p = 0.02, 0.05, 0.1, 0.2, 1\}$ in Table S3. Surprisingly, reducing the number of selected prototypes leads to overall improvements in both $k$-NN accuracy while preserving SIA. We hypothesize that this may be attributed to the reduced input dimensionality of the RF-AE network, which effectively lowers its complexity and introduces additional implicit regularization. Furthermore, selecting only the most representative instances per class may help denoise the training process, thereby enhancing the model's ability to preserve class-relevant features in the embedding space.

For the loss balancing hyperparameter, setting $\lambda = 1$ (i.e., an unconstrained RF-AE) yields relatively high accuracy but results in a substantial decline in supervised structure preservation. This is expected, as unconstrained autoencoders have been shown to poorly capture the underlying data geometry [14, 37]. On the other hand, $\lambda = 0$, which corresponds to the RF-PHATE kernel-based MLP, leads to both lower accuracy and diminished global SIA, offering no improvement over the standard RF-PHATE extension results reported in Table 1.

Across a broad range of hyperparameters—specifically, $\lambda \in \{0.1, 0.01, 0.001\}$ and $N^* \in \{pN_{\text{train}} \mid p = 0.02, 0.05, 0.1, 0.2, 1\}$—RF-AE consistently ranks among the top 3 methods across all metrics in Table 1, highlighting its strong robustness to hyperparameter choices. We note that adding a small geometric constraint to RF-AE improves global supervised structure while still preserving local structure. This finding aligns with the observations of Graving et al. [31], who enhanced $t$-SNE's (unsupervised) global structure by combining it with a VAE.

Fig. S3 illustrates the impact of varying $(\lambda, N^*) \in \{1, 0.1, 0.01, 0.001, 0\} \times \{pN_{\text{train}} \mid p = 0.02, 0.05, 0.1, 0.2, 1\}$ on Sign MNIST (A–K). A smaller number of selected prototypes $N^*$ led to more clearly separated classes and denoised structure, in line with our quantitative findings. When RF-AE is unconstrained ($\lambda = 1$, first column), the resulting embeddings appear more distorted and less structured. This is consistent with recent findings showing that unregularized autoencoders

Table S3: Local ($s = QNX, Trust$) and global ($s = Spear, Pearson$) SIA scores, and test $k$-NN accuracy for RF-AE variants across values of $\lambda$ and $N^*$ (in $\%N_{\text{train}}$). Scores are shown as mean ± std across 20 datasets and 10 repetitions. Each score is compared with baseline models in Table 1, and highlighted only if it ranks among the top three overall. Top three values per metric are highlighted in blue, using underlined bold (first) and bold (second). RF-AE demonstrates strong robustness for $\lambda \in \{0.1, 0.01, 0.001\}$ across varying prototype count, consistently ranking among the top 3 methods. Fewer prototypes improve $k$-NN accuracy while preserving SIA, likely due to implicit regularization and class-level denoising. Extreme $\lambda$ values lead to degraded SIA ($\lambda = 1$) or both SIA and accuracy ($\lambda = 0$).

| | LOCAL SIA | | GLOBAL SIA | | |
| --- | --- | --- | --- | --- | --- |
| | QNX | TRUST | SPEAR | PEARSON | $k$-NN ACC |
| $N^*$ | | | $\lambda = 0$ | | |
| 2% | **0.811 ± 0.026** | **0.822 ± 0.023** | 0.752 ± 0.038 | 0.752 ± 0.040 | **0.839 ± 0.011** |
| 5% | **0.811 ± 0.024** | **0.821 ± 0.022** | 0.752 ± 0.037 | 0.753 ± 0.040 | **0.838 ± 0.010** |
| 10% | **0.812 ± 0.025** | **0.822 ± 0.023** | 0.751 ± 0.038 | 0.752 ± 0.040 | **0.836 ± 0.011** |
| 20% | **0.815 ± 0.024** | **0.822 ± 0.022** | 0.751 ± 0.037 | 0.752 ± 0.040 | **0.833 ± 0.010** |
| 100% | **0.810 ± 0.025** | **0.820 ± 0.022** | 0.750 ± 0.037 | 0.751 ± 0.040 | **0.829 ± 0.010** |
| | | | $\lambda = 0.001$ | | |
| 2% | **0.810 ± 0.024** | **0.824 ± 0.023** | **0.771 ± 0.037** | **0.764 ± 0.040** | **0.857 ± 0.009** |
| 5% | **0.808 ± 0.022** | **0.823 ± 0.022** | **0.768 ± 0.036** | **0.760 ± 0.040** | **0.859 ± 0.008** |
| 10% | **0.809 ± 0.024** | **0.824 ± 0.024** | 0.763 ± 0.037 | 0.757 ± 0.039 | **0.859 ± 0.009** |
| 20% | **0.812 ± 0.025** | **0.826 ± 0.023** | 0.762 ± 0.039 | 0.756 ± 0.042 | **0.855 ± 0.009** |
| 100% | **0.806 ± 0.026** | **0.823 ± 0.024** | 0.756 ± 0.038 | 0.752 ± 0.041 | **0.837 ± 0.012** |
| | | | $\lambda = 0.01$ | | |
| 2% | **0.808 ± 0.024** | **0.822 ± 0.022** | **0.783 ± 0.040** | **0.779 ± 0.042** | **0.859 ± 0.009** |
| 5% | **0.807 ± 0.024** | **0.822 ± 0.021** | **0.782 ± 0.037** | **0.779 ± 0.040** | **0.863 ± 0.008** |
| 10% | **0.809 ± 0.024** | **0.822 ± 0.022** | **0.782 ± 0.041** | **0.779 ± 0.042** | **0.861 ± 0.009** |
| 20% | **0.809 ± 0.024** | **0.822 ± 0.023** | **0.778 ± 0.040** | **0.775 ± 0.040** | **0.860 ± 0.009** |
| 100% | **0.801 ± 0.024** | **0.819 ± 0.023** | **0.773 ± 0.044** | **0.768 ± 0.046** | **0.843 ± 0.012** |
| | | | $\lambda = 0.1$ | | |
| 2% | **0.808 ± 0.023** | **0.822 ± 0.023** | **0.777 ± 0.045** | **0.780 ± 0.043** | **0.862 ± 0.010** |
| 5% | **0.807 ± 0.023** | **0.822 ± 0.021** | **0.778 ± 0.047** | **0.781 ± 0.047** | **0.865 ± 0.008** |
| 10% | **0.808 ± 0.023** | **0.822 ± 0.021** | **0.782 ± 0.060** | **0.784 ± 0.054** | **0.864 ± 0.009** |
| 20% | **0.807 ± 0.025** | **0.820 ± 0.022** | **0.780 ± 0.049** | **0.783 ± 0.049** | **0.861 ± 0.010** |
| 100% | **0.802 ± 0.022** | 0.817 ± 0.024 | **0.780 ± 0.059** | **0.784 ± 0.059** | **0.843 ± 0.012** |
| | | | $\lambda = 1$ | | |
| 2% | **0.808 ± 0.026** | **0.822 ± 0.023** | 0.681 ± 0.113 | 0.681 ± 0.120 | **0.865 ± 0.009** |
| 5% | **0.806 ± 0.024** | **0.820 ± 0.023** | 0.673 ± 0.113 | 0.670 ± 0.109 | **0.867 ± 0.009** |
| 10% | **0.804 ± 0.023** | **0.820 ± 0.022** | 0.689 ± 0.102 | 0.686 ± 0.108 | **0.864 ± 0.009** |
| 20% | **0.804 ± 0.024** | **0.819 ± 0.023** | 0.697 ± 0.118 | 0.694 ± 0.115 | **0.860 ± 0.010** |
| 100% | **0.799 ± 0.023** | 0.812 ± 0.024 | 0.717 ± 0.075 | 0.713 ± 0.075 | 0.808 ± 0.020 |

often fail to produce human-interpretable visualizations that preserve the intrinsic geometry of the data [14, 37]. On the contrary, full geometric constraint ($\lambda = 0$, last column) simply replicates the RF-PHATE embedding, without clear qualitative benefits compared to the default linear kernel extension (Fig. 2). To effectively balance reconstruction and geometric losses, the optimal range for $\lambda$ lies approximately between 0.001—yielding branching structures akin to RF-PHATE but with more pronounced separation—and 0.1, which produces more compact, globular embeddings with enhanced class separability. A similar qualitative assessment can be made for OrganC MNIST in Fig. S4.

From these results, we draw two practical guidelines to help users select suitable hyperparameters for their specific application:

- **Loss balancing parameter $\lambda$:** Values of $\lambda$ in the range $[0.001, 0.1]$ yield comparable scores but differ in qualitative behavior. Lower values (e.g., $\lambda \approx 0.001$) produce branching structures similar to RF-PHATE, enhancing interpretability of inter-class transitions while mitigating the over-compression artifacts seen in RF-PHATE. Higher values (e.g., $\lambda \approx 0.1$) shift the focus toward class separability and expanded within-class structure. We recommend $\lambda \approx 0.001$ for capturing smooth transitions or trajectories, and $\lambda \approx 0.1$ for emphasizing distinct class boundaries and detailed internal structure.

- **Prototype selection $N^*$:** Selecting as few as 2% of training points as prototypes is a good starting point to preserve supervised structure while maximizing class separability. If minimizing inference time is essential, users may further reduce the number of selected prototypes to accelerate computation.

## H.2   Compatibility with other geometric regularizers

Although our RF-PHATE regularizer is the core focus of our paper, we also investigate the performance of RF-AE under alternative geometric constraints to guide users toward potential substitutes. Table S4 reports ablation results using RF-PHATE (ours), UMAP, SUMAP, and RF-UMAP (i.e., UMAP applied to RF-GAP dissimilarities). We fixed the default hyperparameters to $(\lambda, N^*) = (0.01, 0.1N_{\text{train}})$. Across 20 datasets, RF-AE with RF-PHATE consistently achieved the best overall performance. RF-AE with RF-UMAP ranked second, followed by RF-AE with SUMAP and UMAP constraints. On a per-dataset basis, RF-AE (RF-UMAP) was competitive with RF-AE (RF-PHATE) on OrganC MNIST but performed substantially worse on Sign MNIST. These results suggest that RF-AE is especially effective when paired with RF-based kernel methods—particularly RF-PHATE—which already capture the underlying RF geometry. In such cases, the geometric and reconstruction objectives are well aligned, enabling more effective multi-task learning.

Figure S5 shows the OOS visualizations using these four regularizers. On Sign MNIST (Fig. S5a), RF-AE with RF-UMAP still splits same-class clusters, similar to UMAP and SUMAP, though less severely. RF-AE with UMAP or SUMAP attempts to merge same-class fragments, but misalignment with RF-GAP geometry leads to higher class overlap than their parametric counterparts (Fig. S6). On OrganC MNIST (Fig. S5b), which inherently contains less background noise than Sign MNIST, RF-AE with RF-UMAP better highlights anatomical relationships compared to P-UMAP or P-SUMAP, but still shows more noise and overlap than RF-PHATE. This supports the idea that RF-PHATE more effectively captures denoised local and global supervised structure through diffusion, as demonstrated empirically in prior work [4].

In summary, RF-PHATE is a strong default regularizer for RF-AE overall, though alternative RF-based regularizers like RF-UMAP may offer valuable refinements in specific scenarios.

## H.3   SIA performance comparison under different classification importance strategies

To show that RF-AE's performance is not dependent on the choice classification importances $\mathcal{C}_i$ (Section 3.4), we repeated the quantitative analysis from Section 4.1 using two alternative strategies:

- **$k$-NN strategy:** We replaced our baseline ensemble classifier with a standalone $k$-NN model.

- **Aggregate strategy:** Instead of deriving feature importances from the ensemble's accuracy drop, we computed importances independently using each of the three classifiers—$k$-NN, SVM, and MLP—resulting in the following sets:

$$\mathcal{C}^{k\text{-NN}} = \{\mathcal{C}_i^{k\text{-NN}} \mid i = 1, \ldots, D\},$$
$$\mathcal{C}^{\text{SVM}} = \{\mathcal{C}_i^{\text{SVM}} \mid i = 1, \ldots, D\},$$
$$\mathcal{C}^{\text{MLP}} = \{\mathcal{C}_i^{\text{MLP}} \mid i = 1, \ldots, D\}.$$

We then averaged these to obtain an aggregated importance set:

$$\mathcal{C}^{\text{agg}} = \frac{1}{3}\left(\mathcal{C}^{k\text{-NN}} + \mathcal{C}^{\text{SVM}} + \mathcal{C}^{\text{MLP}}\right).$$

Table S5 reports local and global SIA scores for RF-AE and 13 baseline methods using our proposed ensemble classifier (Sections 3.4 and 4.1) as well as the two alternative importance strategies. Overall, RF-AE consistently ranks among the top three methods across all metrics, regardless of the chosen importance strategy. This suggests that RF-AE more effectively preserves the underlying important structure, making it more likely to reflect meaningful feature hierarchies in its embeddings compared to other baselines.

# I   Extended visualizations and quantitative comparisons

## I.1   Visualizations on image data

We present OOS visualization plots and quantitative comparison (Table S6) for all models on Sign MNIST (Fig. S6) and OrganC MNIST (Fig. S7) to support our analysis in Section 4.2.

Table S6 shows the local ($s = QNX, Trust$) and global ($s = Spear, Pearson$) SIA scores, along with test $k$-NN accuracies for RF-AE and 13 baseline methods on the Sign MNIST and OrganC MNIST datasets. Our RF-AE method consistently ranks among the top three across all scores on both datasets.

Fig. S6 presents visualizations of all models for the Sign MNIST (A–K) dataset. RF-AE effectively inherits the global structure of the RF-PHATE embeddings while providing greater detail within class clusters. In contrast, RF-PHATE tends to compress representations within each cluster, which are associated with individual classes. Although OOS embeddings are mostly assigned to their correct ground-truth labels, the local arrangement of these samples on the sub-manifold is not easily visualized in RF-PHATE. RF-AE, however, expands the class clusters, revealing within-class patterns that are obscured in the RF-PHATE plot. For example, the top-right cluster in the RF-AE plot illustrates different ways to represent the letter "C", showing a logical transition between variations based on hand shadowing and orientation. Such nuanced differences are more challenging to detect in RF-PHATE, which compresses these representations into an overly restrictive branch structure. This limitation of RF-PHATE may stem from excessive reliance on the diffusion operator, which overemphasizes global smoothing. Since RF-GAP already captures local-to-global supervised neighborhoods effectively, the additional diffusion applied by RF-PHATE likely diminishes fine-grained local details. Thus, we have demonstrated that RF-AE offers a superior balance for visualizing the local-to-global supervised structure compared to the basic RF-PHATE kernel extension.

P-TSNE is effective at identifying clusters of similar samples but often splits points from the same class into distinct, distant clusters. This appears to result from variations such as background shadowing, which obstruct the important part of the image. Thus, "G" and "H" instances are closer than expected due to similar shadowing. In contrast, RF-AE correctly assigns "G" and "H" instances to their own clusters while dissociating between same-class points with different shadowing, effectively reflecting within-class variations. This demonstrates that P-TSNE is overly sensitive to irrelevant factors, such as background differences, which are unrelated to the underlying labels. Similarly, P-UMAP, P-SUMAP and PACMAP exhibit this sensitivity but produces sparser representations. Despite being a supervised method, P-SUMAP incorporates class labels in a way that artificially clusters same-class points, potentially oversimplifying their intrinsic relationships. CEBRA yields a circular pattern that offers limited utility for qualitative interpretation. CE and SSNP embeddings are distorted. NCA retains decent local and global relationships, but within-class variations and transitions between classes are visually less evident than in regularized RF-AE. Other methods produced noisy embeddings.

For the OrganC MNIST dataset, all models are visualized in Fig. S7. As analyzed in Section 4.2, RF-AE achieves notable improvements over competing methods by enabling finer distinctions between organ types. This is particularly evident in its ability to differentiate the left and right kidneys—whereas other methods tend to merge these classes, RF-AE separates them while maintaining their proximity in the embedding space. This reflects anatomical similarity without losing class identity.

In comparison, RF-PHATE maintains the overall structure but merges certain classes (e.g., left/right kidneys), thereby reducing fine-grained resolution. P-TSNE and P-UMAP recover local structure but yield overlapping clusters due to the lack of supervision, resulting in cluttered embeddings that hinder interpretation. P-SUMAP achieves better class separability than P-TSNE and P-UMAP, but its projections remain difficult to interpret, with elongated structures (e.g., aorta and inferior vena

cava) and compact, overlapping anatomical regions near the center that obscure class boundaries. NCA, PLS-DA, SPCA, and PCA produce noisy visualizations with weak separation of organ types, reflecting limited class-specific representation. Outliers in the CE and SSNP plots suggest overfitting to the training data. PACMAP exhibits broken structures, where organ clusters are artificially split without clear biological meaning. CEBRA displays an artificial circular pattern, while the AE produces distorted visualizations.

Overall, RF-AE preserves the structural integrity of the data while substantially enhancing class separability. These qualitative findings align with the quantitative results in Table S6, where RF-AE outperforms competing methods in both $k$-NN accuracy and local-to-global SIA.

## I.2  Visualizations on audio data

To further demonstrate the modality-agnostic performance of RF-AE, we present quantitative and qualitative comparisons on GTZAN (3-sec) in Table S6 (bottom) and Fig. S8, respectively. Although RF-AE shows slightly lower class separability than models such as CE and SSNP, these methods suffer from strong structural distortions, reflected in their low Global SIA scores. In contrast, RF-AE achieves robust supervised structure preservation across local and global scales while maintaining competitive class separability.

Visually, the class relationships in RF-AE align well with our general understanding of genre similarities and distinctions. For instance, classical, metal, reggae, and hip-hop appear as more "extreme" genres, while disco, rock, country, and blues cluster near the center, reflecting their less distinctive "sound color" and stronger similarities to one another. The proximity of classical and jazz is intuitive, as both often rely on acoustic, traditional instruments. Similarly, metal and rock appear close due to their common reliance on electric guitars and the overlap between subgenres like heavy metal and hard rock. Note that the relatively small sample size and possible biases in label assignment or class-wise sampling may still influence the results. This qualitative analysis is meant to illustrate that RF-AE captures a meaningful and balanced structure in its embeddings, opening the door to further exploration. Future work could examine within- and between-genre variations by coloring points according to key acoustic features.

In contrast, RF-PHATE produces a similar global layout but smooths away important within-class variations, oversimplifying the diversity within each genre. Although CE and SSNP achieve better average class separation, they tend to represent classes as compact, globular clusters. This can be misleading, as it may suggest that musical genres share similar internal structure. In addition, these methods often introduce distortions, elongating some structures while compressing others near the center, which hinders effective visual exploration. PACMAP, P-SUMAP, P-UMAP, and P-TSNE tend to fragment into small clusters, even within genres, making it difficult to observe gradual transitions within and across musical styles. CEBRA again produces its characteristic circular pattern, while the remaining methods yield noisier and less interpretable visualizations.

## J  Semi-supervised training

While we did not experiment with partially labeled data, RF-AE can also be trained in a semi-supervised setting. As described in Section 3.1, our extended RF-GAP definition supports computing proximities between training and out-of-sample points. Thus, on the one hand, assuming $N_L$ labeled points and $N_U$ unlabeled points, for a total training size of $N = N_L + N_U$, we treat unlabeled training samples as "out-of-sample" and compute $N$ proximity vectors of size $N_L$, which are used as input to train the RF-AE network. On the other hand, to generate RF-PHATE embeddings for the full training set, we can rely on the Landmark PHATE algorithm proposed by Moon et al. [3]. First, we construct the RF-GAP kernel matrix of size $N_L \times N_L$ between labeled landmarks and compute their embeddings with PHATE. Then, we project the $N_U$ remaining unlabeled points with the linear landmark extension using their RF-GAP proximities to the labeled points (the landmarks), as in Section 2.2. This provides all the key ingredients to train RF-AE by leveraging both labeled and unlabeled data. We leave this extension for future work.

# K    Runtime comparison

We report training and test runtimes for each model on Sign MNIST (A–K) and OrganC MNIST in Fig. S9. To assess our scalability improvement from our prototype selection, we include results for RF-AE with different prototype percentages, $N^* \in \{0.1N_{\text{train}}, 0.02N_{\text{train}}\}$. We set the geometric weight to its default value $\lambda = 0.01$. During training, RF-AE remains within the same order of magnitude (OoM) as RF-PHATE, P-SUMAP, and NCA, while being one OoM slower than CE and SSNP. At inference, RF-AE is roughly two OoM faster than RF-PHATE and one OoM slower than P-SUMAP. Compared to RF-PHATE, these improvements at inference stem from prototype selection, which avoids the costly computation of proximities to all training points. Combined with our ongoing vectorized and sparse RF-GAP computation, we expect this strategy to substantially narrow, if not eliminate, the runtime gap with other supervised competitors such as P-SUMAP.

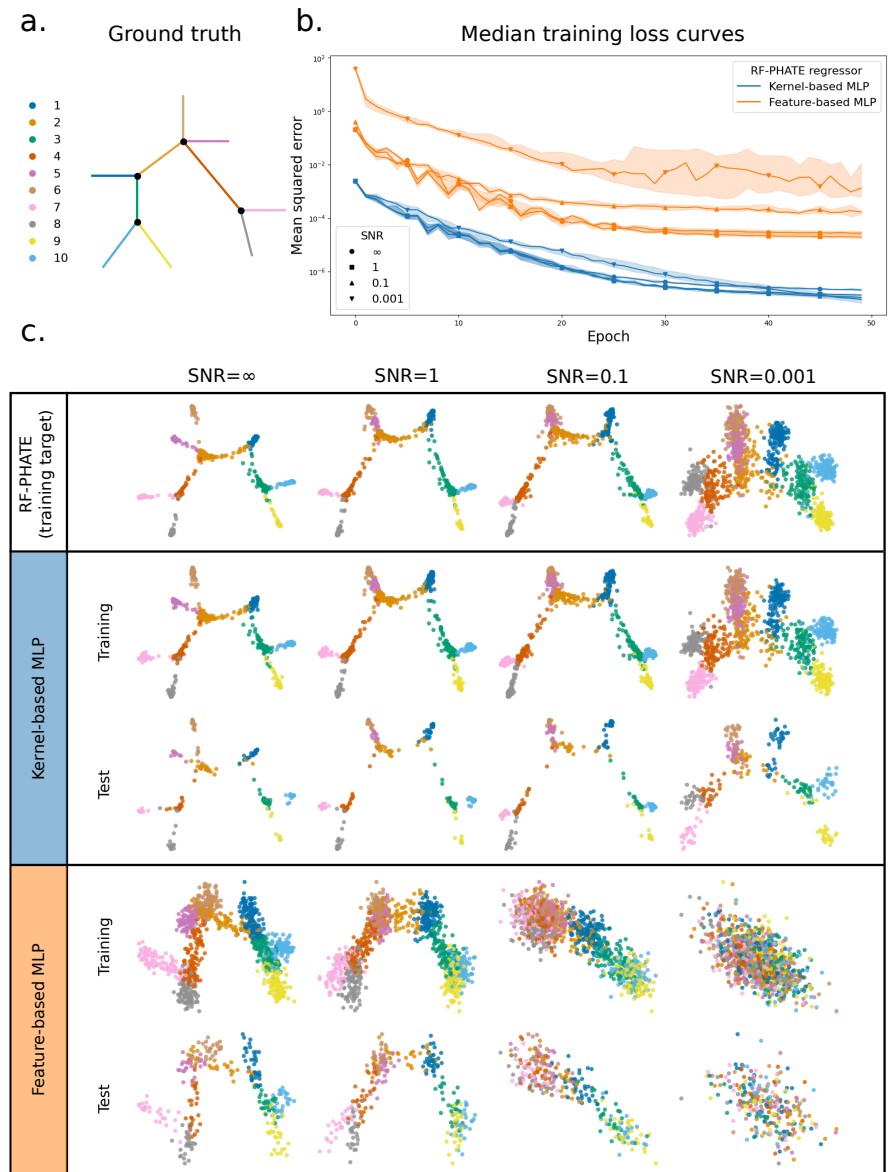

Figure S1: Comparison between the standard feature-based MLP encoder and our proposed RF-GAP kernel-based MLP encoder for regressing onto precomputed RF-PHATE embeddings. **(a)** Ground-truth tree structure with branch labels (see Appendix C). **(b)** Log-scaled median training MSE with $25^{th}$ and $75^{th}$ enclosing percentiles over 50 epochs across 10 repetitions. **(c)** Training RF-PHATE embeddings (top row), followed by training and test embeddings produced by the RF-GAP-based encoder (middle row) and the feature-based encoder (bottom row) after 50 epochs from a single run. The RF-PHATE embeddings closely match the ground-truth structure and provide a strong target for supervised regression. Our kernel-based encoder remains effective even under high noise levels (e.g., SNR = 0.001), converging more quickly and producing well-structured embeddings with better generalization. In contrast, the feature-based MLP exhibits increasing training loss and disorganized embeddings as noise increases, and often fails to recover meaningful structure even in low-noise settings (SNR = $\infty$, 1), demonstrating the superior robustness of our kernel-based encoders.

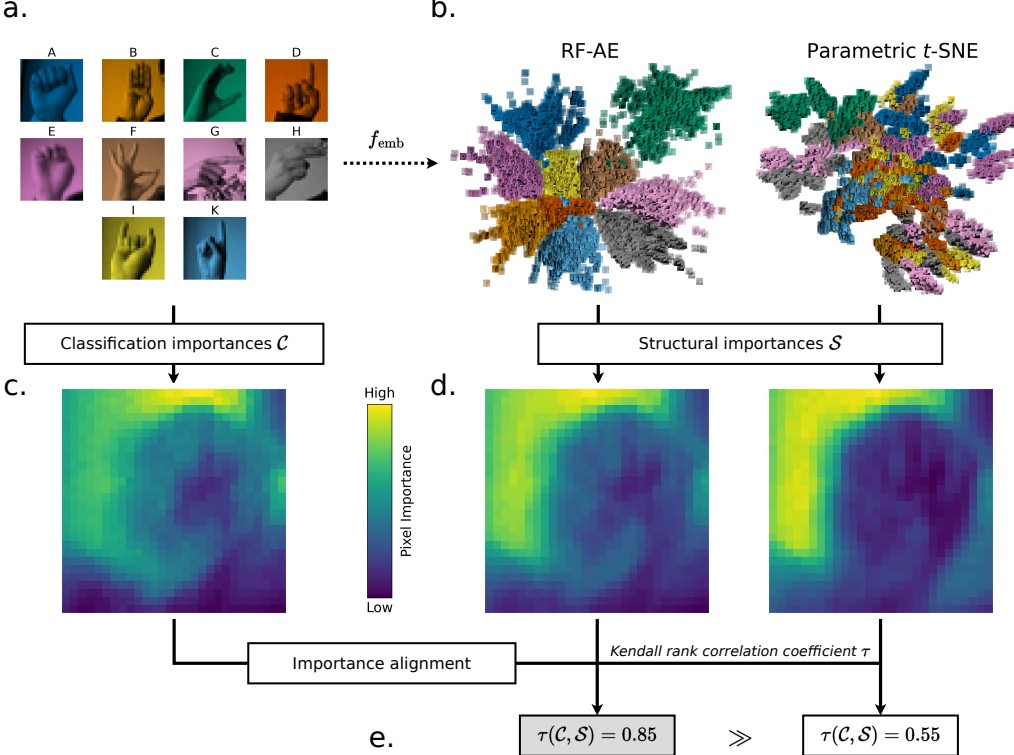

Figure S2: Illustration of the structural importance alignment (SIA) score defined in Section 3.4 for evaluating supervised out-of-sample (OOS) embedding fit. **a.** Random class samples from the high-dimensional Sign MNIST (A–K) dataset. **b.** 2D embeddings of training and test (OOS) points from RF-AE (left) and P-TSNE (right), based on a stratified 80/20 random split. Samples are shown with their original images, color-tinted by label; training samples appear with reduced opacity. **c.** Pixel-level classification importances from the ensemble baseline classifier (Section 3.4, Appendix E.3), normalized to $[0, 1]$. **d.** Pixel-level local structure importances ($s = $ *Trust*) from OOS RF-AE (left) and P-TSNE (right), also normalized. **e.** Local SIA scores computed as the Kendall $\tau$ correlation between (c) and (d): RF-AE achieves higher alignment (0.85) than P-TSNE (0.55), suppressing background pixels and focusing on class-relevant regions.

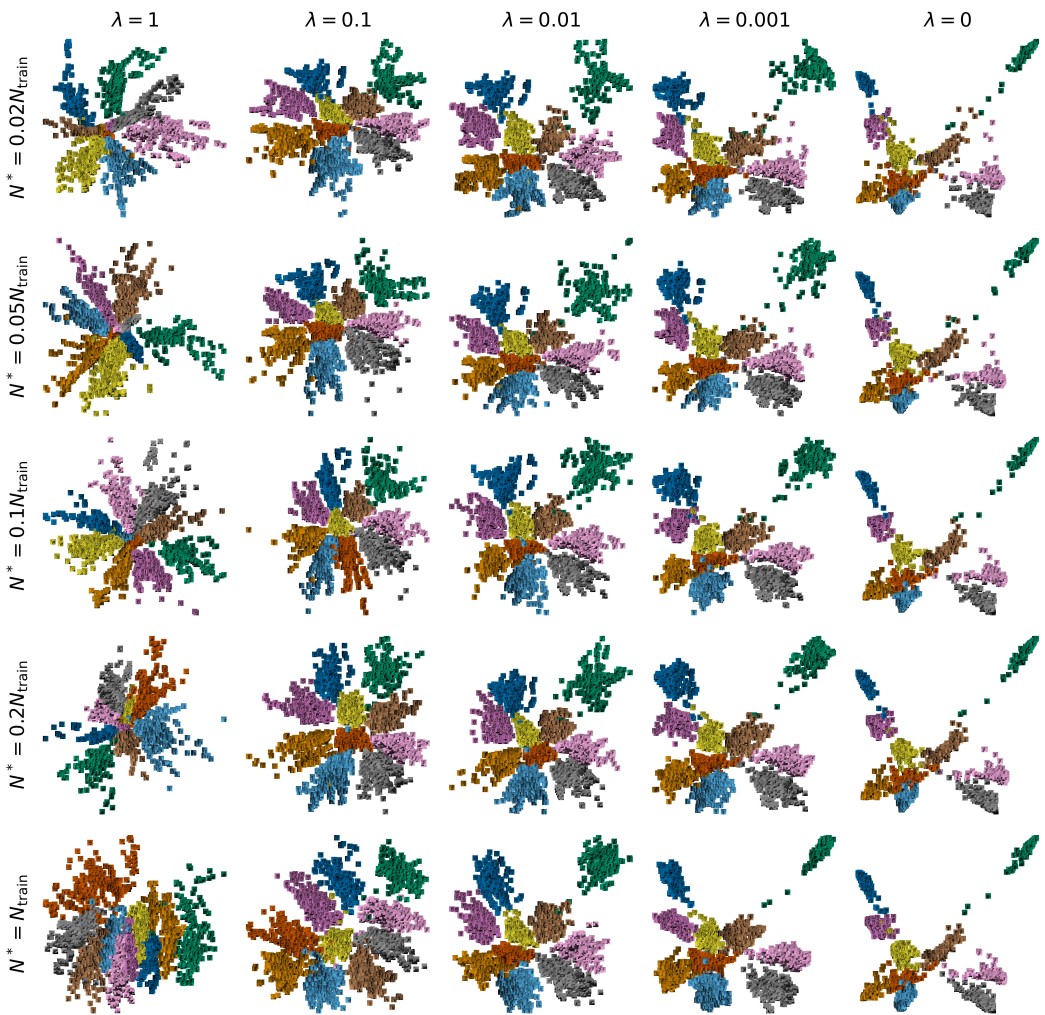

Figure S3: RF-AE test embeddings on Sign MNIST (A–K) for various $(\lambda, N^*)$ configurations, where $\lambda$ decreases column-wise from 1 (unconstrained RF-AE) to 0 (RF-PHATE kernel-based MLP extension), and $N^*$ increases row-wise from 2% to 100% of the training set size. Samples are shown with their original images, color-tinted by label (see Fig. 2 for the legend).

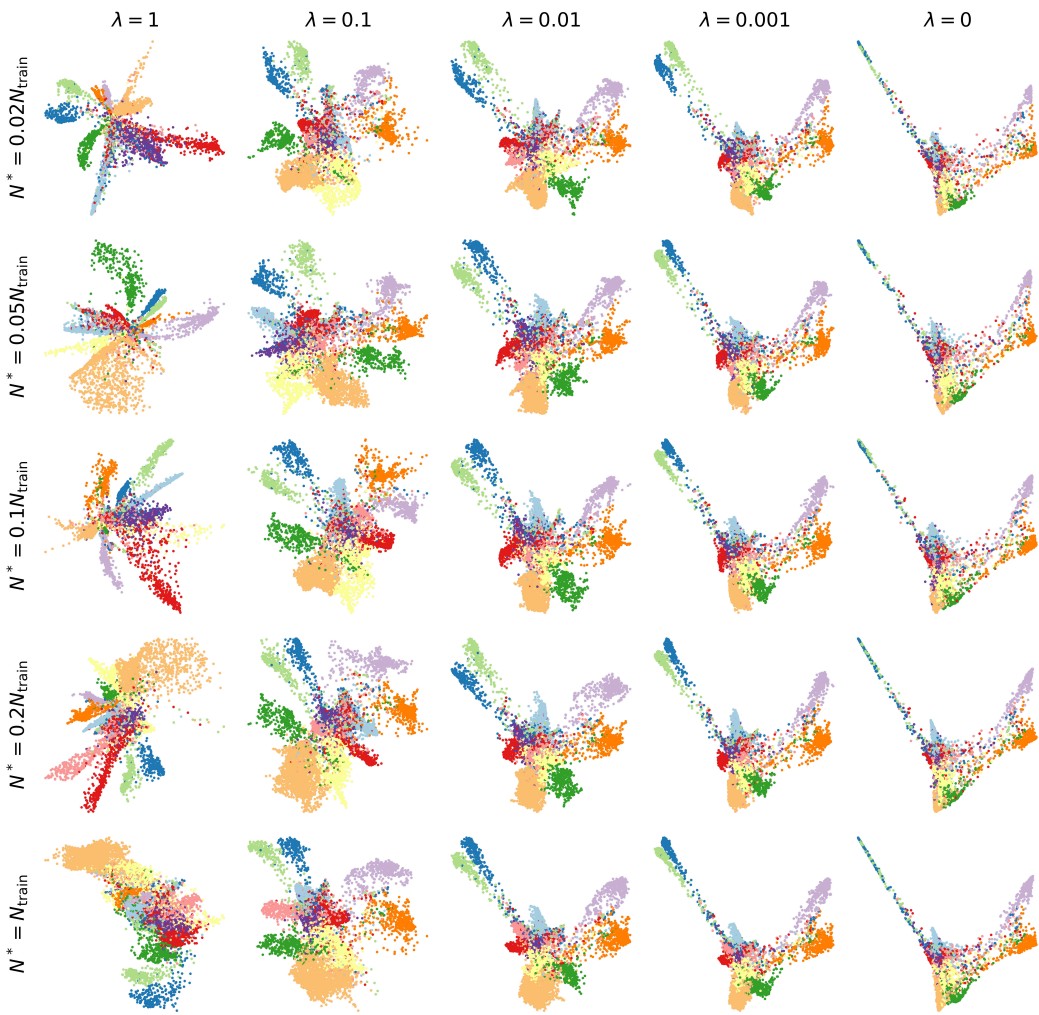

Figure S4: RF-AE test embeddings on OrganC MNIST for various $(\lambda, N^*)$ configurations, where $\lambda$ decreases column-wise from 1 (unconstrained RF-AE) to 0 (RF-PHATE kernel-based MLP extension), and $N^*$ increases row-wise from 2% to 100% of the training set size. Points are colored by their ground-truth labels (see Fig. 2 for the legend).

Table S4: Local ($s = QNX$, $Trust$) and global ($s = Spear$, $Pearson$) SIA scores, along with test $k$-NN accuracies for our RF-AE method using four different geometric regularizers: RF-PHATE (ours), RF-UMAP, UMAP and SUMAP. Scores are shown as mean $\pm$ std across 10 repetitions on Sign MNIST (top), OrganC MNIST (middle), and over 20 datasets (bottom). Refer to Table S2 for a summary of the 20 datasets. Each score is compared with baseline models in Tables 1 and S6, and highlighted only if it ranks among the top three overall. Top three values per metric are highlighted in blue, using underlined bold (first) and bold (second).

| | LOCAL SIA | | GLOBAL SIA | | |
| | QNX | TRUST | SPEAR | PEARSON | $k$-NN ACC |
|---|---|---|---|---|---|
| GEO. REG. | | | SIGN MNIST | | |
| RF-PHATE | **0.819 ± 0.006** | 0.848 ± 0.006 | **0.700 ± 0.109** | **0.681 ± 0.135** | **0.988 ± 0.003** |
| RF-UMAP | 0.732 ± 0.008 | 0.717 ± 0.009 | 0.624 ± 0.043 | 0.600 ± 0.040 | 0.936 ± 0.016 |
| UMAP | 0.642 ± 0.012 | 0.544 ± 0.015 | 0.319 ± 0.048 | 0.235 ± 0.070 | 0.745 ± 0.021 |
| SUMAP | 0.668 ± 0.012 | 0.594 ± 0.015 | 0.461 ± 0.069 | 0.414 ± 0.097 | 0.863 ± 0.013 |
| | | | ORGANC MNIST | | |
| RF-PHATE | 0.890 ± 0.007 | **0.929 ± 0.006** | **0.901 ± 0.013** | **0.898 ± 0.012** | **0.766 ± 0.004** |
| RF-UMAP | 0.889 ± 0.006 | **0.924 ± 0.005** | **0.936 ± 0.004** | **0.933 ± 0.005** | **0.701 ± 0.004** |
| UMAP | 0.883 ± 0.007 | 0.907 ± 0.006 | 0.871 ± 0.006 | 0.869 ± 0.006 | 0.576 ± 0.007 |
| SUMAP | 0.875 ± 0.008 | 0.909 ± 0.006 | 0.888 ± 0.009 | 0.875 ± 0.008 | **0.740 ± 0.013** |
| | | | 20 DATASETS | | |
| RF-PHATE | **0.809 ± 0.024** | **0.822 ± 0.022** | **0.782 ± 0.041** | **0.779 ± 0.042** | **0.861 ± 0.009** |
| RF-UMAP | **0.798 ± 0.024** | 0.806 ± 0.022 | **0.773 ± 0.031** | **0.768 ± 0.032** | **0.832 ± 0.012** |
| UMAP | 0.782 ± 0.025 | 0.762 ± 0.029 | 0.683 ± 0.036 | 0.674 ± 0.038 | 0.729 ± 0.024 |
| SUMAP | 0.791 ± 0.024 | 0.788 ± 0.024 | 0.669 ± 0.050 | 0.669 ± 0.048 | **0.817 ± 0.017** |

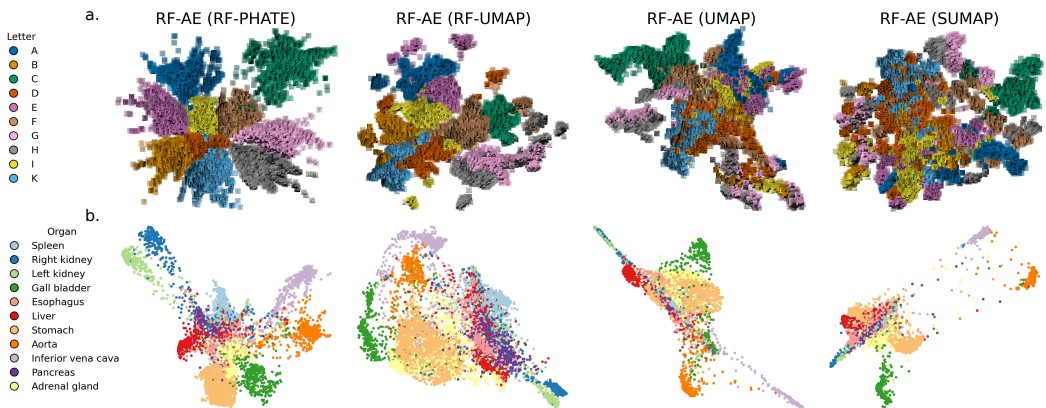

Figure S5: OOS visualization using RF-AE with four different geometric regularizers: RF-PHATE (ours, far left), RF-UMAP (center left), UMAP (center right) and SUMAP (far right). We set the geometric and reconstruction weights to their default values $(\lambda, N^*) = (0.01, 0.1 N_{\text{train}})$. **a.** Sign MNIST (A–K): Samples are shown with their original images, color-tinted by label. Training images are shown with reduced opacity. RF-AE with RF-UMAP still fragments same-class points, reflecting the same weaknesses as (un)supervised UMAP (Fig. S6). RF-AE with UMAP or SUMAP attempts to merge same-class fragments but misalignment with RF-GAP geometry leads to greater class overlap than their parametric baselines. **b.** OrganC MNIST: Test points are color-coded by label. Training points are omitted for clarity. RF-AE with RF-UMAP performs better, producing a structure closer to RF-AE with RF-PHATE. The reduced artifact level (e.g., less background noise) facilitates clustering of same-class points. Still, RF-UMAP remains slightly noisier than RF-PHATE, with higher class overlap, and RF-AE with UMAP or SUMAP shows no improvement over their parametric counterparts in Fig. S7.

Table S5: Local ($s = QNX, Trust$) and global ($s = Spear, Pearson$) SIA scores for RF-AE and 13 baseline methods, computed using three strategies: our default ensemble importances suggested in Section 3.4 (top), $k$-NN-based classification importances (middle) and aggregated importances averaged over $k$-NN, SVM, and MLP classifiers (bottom). Scores are reported as mean $\pm$ standard deviation across 20 datasets and 10 repetitions. In general, RF-AE outperforms other models in both local and global SIA, regardless of the importance strategy. Top three values per metric are highlighted in blue, using underlined bold (first) and bold (second). Supervised methods are marked by an asterisk.

| | LOCAL SIA | | GLOBAL SIA | |
|---|---|---|---|---|
| | QNX | TRUST | SPEAR | PEARSON |
| ENSEMBLE IMPORTANCES | | | | |
| RF-AE* | **0.809 ± 0.024** | 0.822 ± 0.022 | **0.782 ± 0.041** | **0.779 ± 0.042** |
| RF-PHATE* | 0.798 ± 0.025 | **0.825 ± 0.023** | 0.748 ± 0.038 | 0.750 ± 0.040 |
| SSNP* | 0.760 ± 0.047 | 0.772 ± 0.045 | 0.685 ± 0.089 | 0.694 ± 0.080 |
| P-SUMAP* | 0.756 ± 0.028 | 0.768 ± 0.025 | 0.647 ± 0.048 | 0.647 ± 0.048 |
| CE* | 0.795 ± 0.050 | 0.818 ± 0.051 | 0.765 ± 0.051 | **0.763 ± 0.054** |
| NCA* | **0.808 ± 0.027** | 0.805 ± 0.025 | **0.771 ± 0.032** | 0.759 ± 0.033 |
| PACMAP | 0.749 ± 0.026 | 0.758 ± 0.025 | 0.688 ± 0.029 | 0.688 ± 0.029 |
| P-TSNE | 0.743 ± 0.028 | 0.747 ± 0.028 | 0.684 ± 0.036 | 0.666 ± 0.038 |
| AE | 0.744 ± 0.027 | 0.751 ± 0.029 | 0.695 ± 0.044 | 0.655 ± 0.053 |
| P-UMAP | 0.757 ± 0.027 | 0.744 ± 0.028 | 0.674 ± 0.035 | 0.657 ± 0.038 |
| SPCA* | 0.767 ± 0.026 | 0.759 ± 0.030 | 0.741 ± 0.031 | 0.738 ± 0.032 |
| PLS-DA* | 0.715 ± 0.026 | 0.708 ± 0.028 | 0.659 ± 0.027 | 0.639 ± 0.028 |
| CEBRA* | 0.780 ± 0.045 | 0.775 ± 0.050 | 0.735 ± 0.062 | 0.728 ± 0.068 |
| PCA | 0.745 ± 0.027 | 0.742 ± 0.026 | 0.733 ± 0.027 | 0.727 ± 0.028 |
| STANDALONE $k$-NN IMPORTANCES | | | | |
| RF-AE* | **0.835 ± 0.021** | 0.832 ± 0.021 | **0.784 ± 0.036** | **0.788 ± 0.038** |
| RF-PHATE* | **0.834 ± 0.024** | **0.836 ± 0.023** | 0.750 ± 0.035 | 0.760 ± 0.040 |
| SSNP* | 0.780 ± 0.050 | 0.779 ± 0.046 | 0.681 ± 0.094 | 0.690 ± 0.084 |
| P-SUMAP* | 0.780 ± 0.026 | 0.788 ± 0.023 | 0.666 ± 0.049 | 0.666 ± 0.049 |
| CE* | 0.829 ± 0.050 | 0.821 ± 0.048 | 0.763 ± 0.051 | 0.760 ± 0.050 |
| NCA* | 0.826 ± 0.022 | 0.811 ± 0.025 | **0.774 ± 0.032** | **0.761 ± 0.031** |
| PACMAP | 0.771 ± 0.023 | 0.777 ± 0.022 | 0.708 ± 0.027 | 0.711 ± 0.028 |
| P-TSNE | 0.766 ± 0.025 | 0.767 ± 0.025 | 0.702 ± 0.030 | 0.683 ± 0.035 |
| AE | 0.762 ± 0.025 | 0.769 ± 0.025 | 0.709 ± 0.046 | 0.668 ± 0.054 |
| P-UMAP | 0.777 ± 0.027 | 0.762 ± 0.025 | 0.695 ± 0.030 | 0.676 ± 0.037 |
| SPCA* | 0.785 ± 0.024 | 0.777 ± 0.026 | 0.753 ± 0.026 | 0.749 ± 0.026 |
| PLS-DA* | 0.724 ± 0.022 | 0.714 ± 0.022 | 0.654 ± 0.023 | 0.634 ± 0.025 |
| CEBRA* | 0.806 ± 0.046 | 0.784 ± 0.049 | 0.736 ± 0.058 | 0.731 ± 0.065 |
| PCA | 0.755 ± 0.023 | 0.752 ± 0.022 | 0.741 ± 0.023 | 0.736 ± 0.024 |
| AGGREGATED IMPORTANCES | | | | |
| RF-AE* | **0.802 ± 0.045** | 0.812 ± 0.044 | **0.778 ± 0.056** | **0.778 ± 0.056** |
| RF-PHATE* | 0.793 ± 0.040 | **0.820 ± 0.043** | 0.747 ± 0.051 | 0.752 ± 0.052 |
| SSNP* | 0.764 ± 0.060 | 0.770 ± 0.056 | 0.685 ± 0.100 | 0.695 ± 0.092 |
| P-SUMAP* | 0.757 ± 0.046 | 0.767 ± 0.044 | 0.647 ± 0.065 | 0.646 ± 0.063 |
| CE* | 0.798 ± 0.062 | **0.819 ± 0.063** | 0.771 ± 0.068 | **0.772 ± 0.067** |
| NCA* | **0.812 ± 0.045** | 0.804 ± 0.046 | **0.774 ± 0.048** | 0.762 ± 0.049 |
| PACMAP | 0.749 ± 0.046 | 0.758 ± 0.044 | 0.688 ± 0.044 | 0.690 ± 0.046 |
| P-TSNE | 0.744 ± 0.044 | 0.747 ± 0.044 | 0.684 ± 0.048 | 0.667 ± 0.051 |
| AE | 0.745 ± 0.043 | 0.750 ± 0.045 | 0.695 ± 0.061 | 0.655 ± 0.066 |
| P-UMAP | 0.760 ± 0.047 | 0.744 ± 0.047 | 0.674 ± 0.048 | 0.657 ± 0.054 |
| SPCA* | 0.770 ± 0.042 | 0.761 ± 0.047 | 0.742 ± 0.045 | 0.739 ± 0.046 |
| PLS-DA* | 0.717 ± 0.043 | 0.710 ± 0.044 | 0.664 ± 0.039 | 0.643 ± 0.040 |
| CEBRA* | 0.782 ± 0.063 | 0.778 ± 0.068 | 0.739 ± 0.073 | 0.733 ± 0.079 |
| PCA | 0.746 ± 0.045 | 0.743 ± 0.044 | 0.734 ± 0.042 | 0.729 ± 0.043 |

Table S6: Local ($s = QNX$, $Trust$) and global ($s = Spear$, $Pearson$) SIA scores, along with test $k$-NN accuracies for our RF-AE method and 13 baselines. Scores are shown as mean ± std across 10 repetitions on Sign MNIST (top), OrganC MNIST (middle) and GTZAN (bottom) (see Table S2 for a summary of the datasets). Top three values per metric are highlighted in blue, using underlined bold (first) and bold (second). Supervised methods are marked by an asterisk.

| | LOCAL SIA | | GLOBAL SIA | | |
| | QNX | TRUST | SPEAR | PEARSON | $k$-NN ACC |
|---|---|---|---|---|---|
| | | | SIGN MNIST | | |
| RF-AE* | **0.819 ± 0.006** | 0.848 ± 0.006 | **0.700 ± 0.109** | **0.681 ± 0.135** | **0.988 ± 0.003** |
| RF-PHATE* | **0.817 ± 0.009** | **0.854 ± 0.011** | 0.571 ± 0.099 | 0.434 ± 0.149 | 0.976 ± 0.004 |
| SSNP* | 0.139 ± 0.401 | 0.249 ± 0.381 | 0.174 ± 0.391 | 0.414 ± 0.219 | 0.189 ± 0.258 |
| P-SUMAP* | 0.700 ± 0.010 | 0.618 ± 0.009 | 0.449 ± 0.079 | 0.401 ± 0.103 | 0.967 ± 0.004 |
| CE* | 0.620 ± 0.408 | 0.627 ± 0.418 | 0.695 ± 0.135 | 0.646 ± 0.184 | 0.464 ± 0.179 |
| NCA* | 0.793 ± 0.013 | **0.873 ± 0.012** | 0.596 ± 0.088 | 0.523 ± 0.110 | **0.984 ± 0.002** |
| PACMAP | 0.718 ± 0.007 | 0.616 ± 0.008 | 0.402 ± 0.026 | 0.382 ± 0.029 | 0.930 ± 0.005 |
| P-TSNE | 0.689 ± 0.010 | 0.535 ± 0.021 | 0.304 ± 0.050 | 0.210 ± 0.084 | 0.806 ± 0.032 |
| AE | 0.668 ± 0.019 | 0.625 ± 0.046 | 0.403 ± 0.165 | 0.361 ± 0.181 | 0.524 ± 0.131 |
| P-UMAP | 0.665 ± 0.012 | 0.551 ± 0.011 | 0.304 ± 0.042 | 0.223 ± 0.064 | 0.787 ± 0.026 |
| SPCA* | 0.676 ± 0.005 | 0.598 ± 0.009 | 0.552 ± 0.011 | 0.519 ± 0.012 | 0.479 ± 0.009 |
| PLS-DA* | 0.740 ± 0.008 | 0.729 ± 0.009 | **0.737 ± 0.011** | **0.735 ± 0.012** | 0.357 ± 0.008 |
| CEBRA* | 0.742 ± 0.064 | 0.744 ± 0.132 | 0.586 ± 0.129 | 0.564 ± 0.149 | 0.430 ± 0.091 |
| PCA | 0.660 ± 0.011 | 0.588 ± 0.015 | 0.576 ± 0.013 | 0.589 ± 0.012 | 0.314 ± 0.006 |
| | | | ORGANC MNIST | | |
| RF-AE* | 0.890 ± 0.007 | **0.929 ± 0.006** | **0.901 ± 0.013** | **0.898 ± 0.012** | **0.766 ± 0.004** |
| RF-PHATE* | 0.892 ± 0.007 | 0.912 ± 0.006 | **0.898 ± 0.009** | **0.896 ± 0.012** | **0.654 ± 0.008** |
| SSNP* | 0.871 ± 0.028 | 0.906 ± 0.019 | 0.773 ± 0.358 | 0.784 ± 0.096 | 0.636 ± 0.154 |
| P-SUMAP* | 0.873 ± 0.006 | 0.898 ± 0.006 | 0.886 ± 0.006 | 0.875 ± 0.006 | 0.618 ± 0.018 |
| CE* | 0.870 ± 0.022 | 0.887 ± 0.024 | 0.854 ± 0.076 | 0.846 ± 0.073 | 0.570 ± 0.193 |
| NCA* | **0.892 ± 0.006** | 0.896 ± 0.005 | 0.870 ± 0.005 | 0.868 ± 0.005 | 0.524 ± 0.000 |
| PACMAP | 0.881 ± 0.007 | 0.902 ± 0.006 | 0.893 ± 0.007 | 0.893 ± 0.006 | 0.632 ± 0.009 |
| P-TSNE | 0.867 ± 0.006 | 0.892 ± 0.005 | 0.874 ± 0.005 | 0.871 ± 0.005 | 0.474 ± 0.003 |
| AE | 0.875 ± 0.006 | 0.899 ± 0.005 | 0.873 ± 0.011 | 0.834 ± 0.022 | 0.563 ± 0.014 |
| P-UMAP | 0.881 ± 0.006 | 0.898 ± 0.004 | 0.870 ± 0.005 | 0.868 ± 0.005 | 0.475 ± 0.005 |
| SPCA* | **0.916 ± 0.005** | **0.926 ± 0.005** | 0.895 ± 0.005 | 0.886 ± 0.005 | 0.429 ± 0.000 |
| PLS-DA* | 0.866 ± 0.006 | 0.869 ± 0.005 | 0.860 ± 0.005 | 0.859 ± 0.005 | 0.358 ± 0.000 |
| CEBRA* | 0.858 ± 0.033 | 0.881 ± 0.032 | 0.872 ± 0.030 | 0.862 ± 0.027 | 0.358 ± 0.034 |
| PCA | 0.861 ± 0.005 | 0.879 ± 0.005 | 0.865 ± 0.005 | 0.861 ± 0.005 | 0.414 ± 0.000 |
| | | | GTZAN (3-SEC) | | |
| RF-AE* | **0.956 ± 0.007** | **0.946 ± 0.005** | **0.935 ± 0.011** | **0.914 ± 0.008** | 0.688 ± 0.005 |
| RF-PHATE* | **0.954 ± 0.005** | 0.943 ± 0.008 | **0.912 ± 0.015** | **0.907 ± 0.014** | 0.568 ± 0.010 |
| SSNP* | 0.940 ± 0.006 | 0.931 ± 0.013 | 0.788 ± 0.056 | 0.778 ± 0.101 | **0.786 ± 0.005** |
| CE* | 0.951 ± 0.008 | 0.931 ± 0.016 | 0.807 ± 0.046 | 0.747 ± 0.049 | **0.713 ± 0.012** |
| P-SUMAP* | 0.934 ± 0.005 | 0.937 ± 0.005 | 0.648 ± 0.010 | 0.638 ± 0.022 | 0.696 ± 0.007 |
| NCA* | 0.949 ± 0.005 | 0.925 ± 0.007 | 0.788 ± 0.018 | 0.824 ± 0.022 | 0.518 ± 0.006 |
| PACMAP | 0.942 ± 0.007 | **0.946 ± 0.004** | 0.706 ± 0.014 | 0.692 ± 0.017 | 0.644 ± 0.009 |
| P-TSNE | 0.950 ± 0.005 | 0.941 ± 0.005 | 0.777 ± 0.017 | 0.787 ± 0.018 | 0.519 ± 0.007 |
| AE | 0.939 ± 0.009 | 0.939 ± 0.006 | 0.809 ± 0.035 | 0.847 ± 0.045 | 0.487 ± 0.009 |
| P-UMAP | 0.949 ± 0.007 | 0.937 ± 0.009 | 0.723 ± 0.012 | 0.704 ± 0.016 | 0.493 ± 0.040 |
| SPCA* | 0.948 ± 0.003 | 0.932 ± 0.008 | 0.894 ± 0.007 | 0.900 ± 0.008 | 0.417 ± 0.009 |
| CEBRA | 0.839 ± 0.058 | 0.841 ± 0.055 | 0.786 ± 0.042 | 0.779 ± 0.040 | 0.309 ± 0.020 |
| PLS-DA* | 0.848 ± 0.015 | 0.817 ± 0.016 | 0.764 ± 0.017 | 0.706 ± 0.015 | 0.398 ± 0.006 |
| PCA | 0.943 ± 0.006 | 0.926 ± 0.008 | 0.887 ± 0.009 | 0.889 ± 0.008 | 0.404 ± 0.005 |

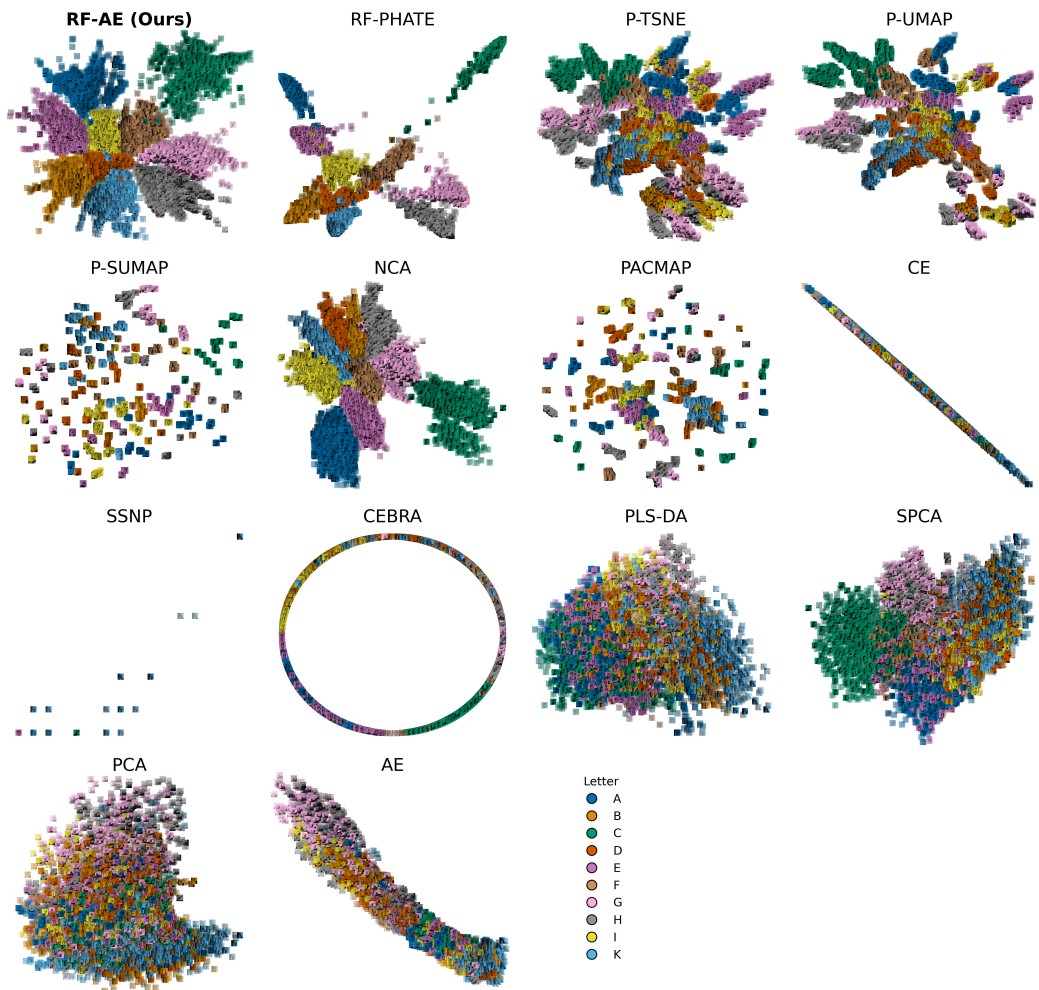

Figure S6: Visualization of the Sign MNIST (A–K) dataset (Table S2) using 14 dimensionality reduction methods. Training and test samples are shown with their original images, color-tinted by label; training samples appear with reduced opacity. See Appendix I.1 for a full qualitative analysis.

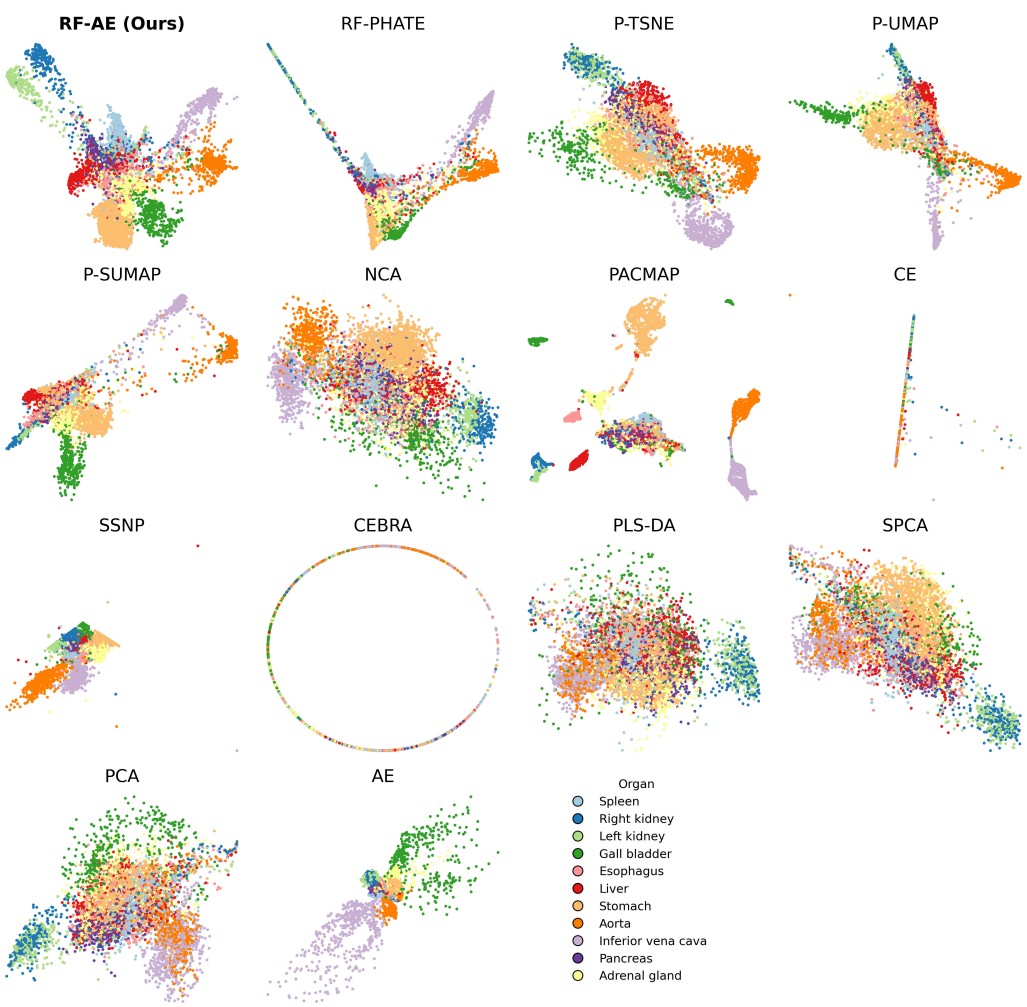

Figure S7: Visualization of the OrganC MNIST dataset (Table S2) using 14 dimensionality reduction methods. Test points are shown as color-coded circles based on their labels. Training points are omitted for clarity. Refer to Appendix I.1 for a full qualitative analysis.

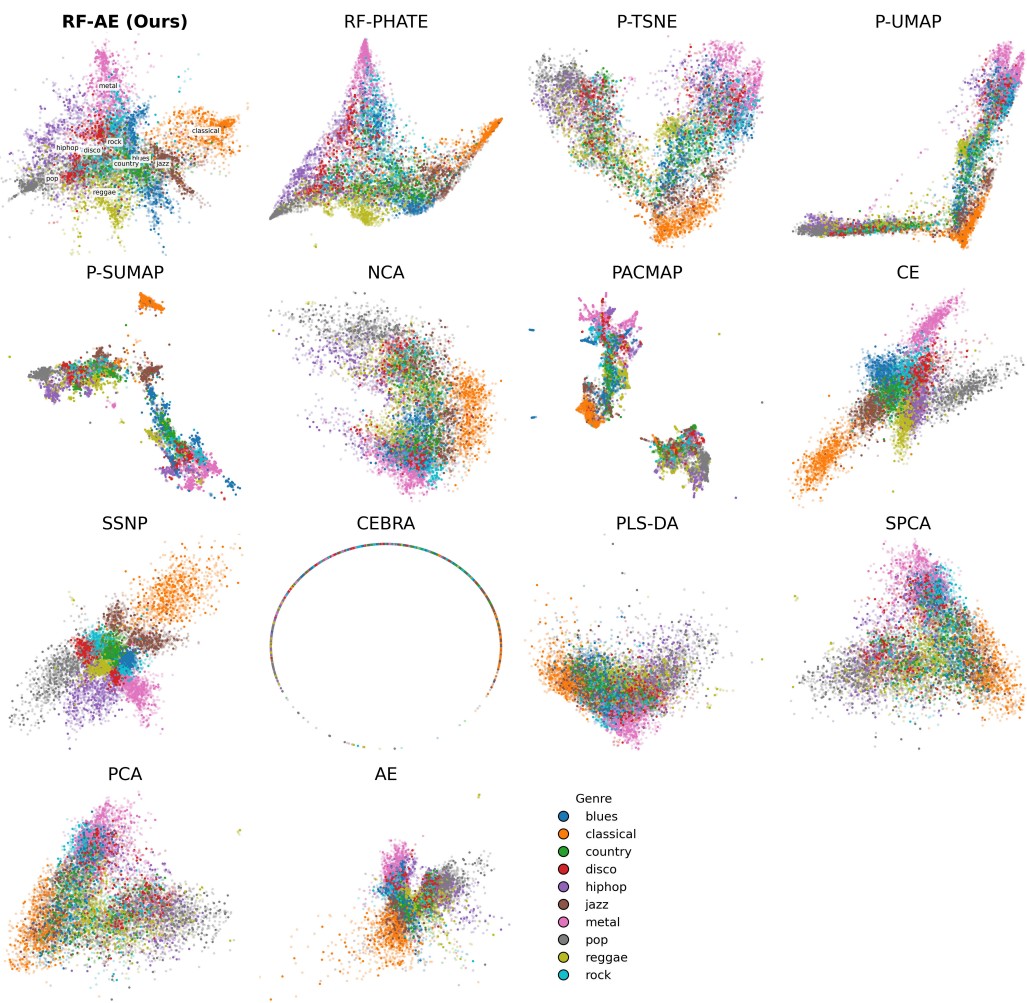

Figure S8: Visualization of the GTZAN (3-sec) dataset (Table S2) using 14 dimensionality reduction methods. Training and test points are shown as color-coded circles based on their labels. Training samples appear with reduced opacity. Refer to Appendix I.2 for a full qualitative analysis.

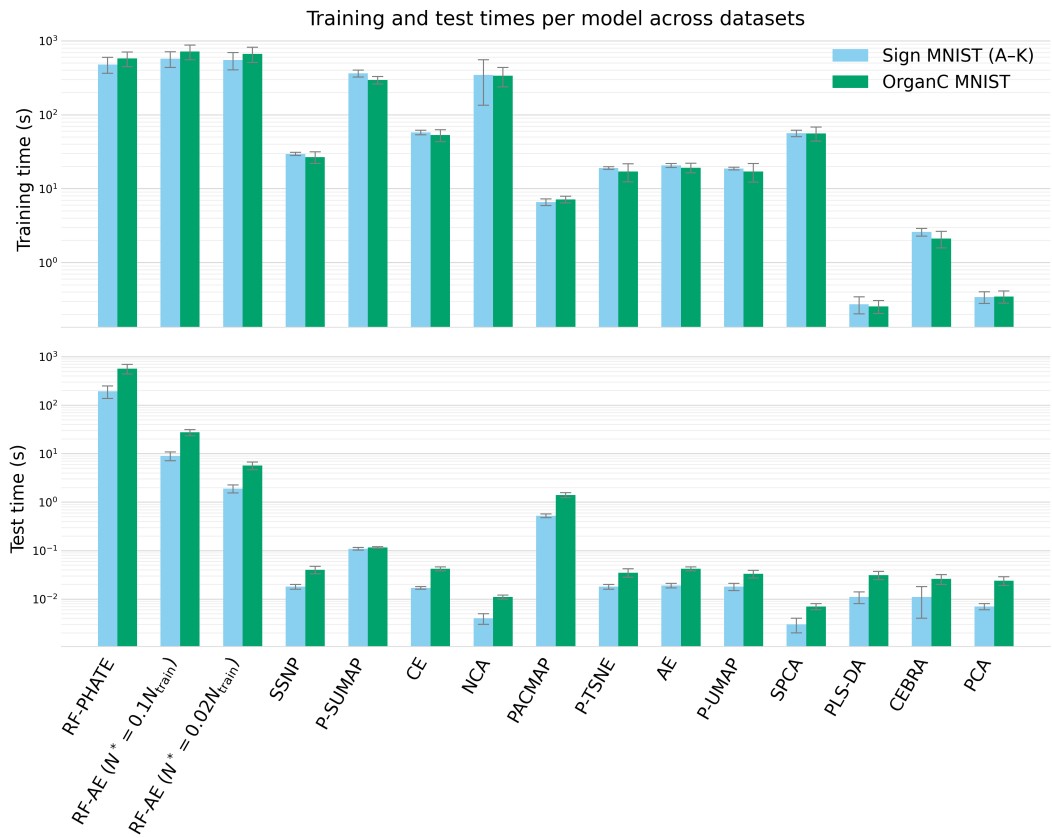

Figure S9: Average training (top) and test (bottom) computation times per model across 10 repetitions on Sign MNIST (blue) and OrganC MNIST (green). Standard deviations are displayed as error bars.

