# OpenReview forum: "Random Forest Autoencoders for Guided Representation Learning"
_NeurIPS.cc/2025/Conference — NeurIPS 2025 poster_

### Official Review · Reviewer_mQiQ · 2025-06-30

**Clarity:** 3
**Significance:** 3
**Originality:** 3
**Rating:** 4
**Confidence:** 3

**Summary:**

## Summary

The paper presents RF-AE, a novel framework for supervised dimension reduction method that addresses the limitations of the existing RF-PHATE method. It provides an explicit mapping function for out-of-sample extension, which is useful for handling large datasets and new data points. RF-AE combines autoencoders with the strengths of random forests and the geometric insights of RF-PHATE. It provides experiment results from comparing RF-AE with existing dimension reduction methods.

**Questions:**

1. How is new data defined?
2. How does RF-AE compare to UMAP in terms of speed and scalability?
3. How can RF-AE incorporate task-specific insights?

**Ethical Concerns:**

["NO or VERY MINOR ethics concerns only"]

**Final Justification:**

Most of my questions have been answered. Given that my original rating was already positive, I will be maintaining it.

**Limitations:**

yes

**Quality:**

3

**Strengths And Weaknesses:**

## Strength

S1. This paper focuses on developing new algorithms for supervised dimension reduction, which is a timely research question. The proposed method is sound and novel.

S2. The experimental design is comprehensive, using 20 datasets and comparing against popular dimension reduction algorithms such as tSNE and UMAP. The quantitative results look promising.

S3. The paper is well-organized and easy to follow.

## Major Weakness

W1. One motivation for RF-AE is to accommodate new observations in during dimension reduction (L29). For example, the abstract and introduction highlight that supervised approaches like RF-PHATE can capture task-specific insights with auxiliary labels. However, section 4 doesn't seem to utilize any task-specific insights or expert labels.

W2. One major benefit for UMAP is its speed and scalability. It is unclear how RF-AE compares to UMAP in these aspects.

## Minor Weakness

M1. How is new data defined? New data from the same distribution of the existing data, or totally out of distribution data?

M2. The term "data visualization" in this paper can lead to confusion. This paper mainly focuses on dimension reduction instead of data visualization. I recommend changing it to dimension reduction.

---

> ### Author Rebuttal · Authors · 2025-07-30
>
> Thank you for your highly relevant remarks and overall positive review.
>
> **Re W1.** The term *expert labels* refers to annotations provided by experts and assigned to each data point in the training set. The training set used in Section 4 is labeled, meaning that each instance is associated with a specific class. Labels are necessary for supervised learning, as they enable us to train a Random Forest and capture task-specific structure in 2D through our RF-AE framework. I.e., the expert labels are used to train the random forest, from which the RF-GAP proximities are extracted. Previous work has extensively shown that the RF-GAP proximities encode the relationships between points, emphasizing the variables that are relevant for the supervised task (e.g. Rhodes et al. [1]). Thus all of the supervised methods in Section 4 were trained using using expert labels, while unsupervised methods operated without labels. For a detailed description of the data sets, including the number of classes in each, please refer to Appendix F.
>
>
> **Re W2.** Thank you for pointing this out. We agree that one of UMAP’s key strengths is its speed and scalability, particularly in unsupervised visualization tasks. However, RF-AE is designed for a different goal: producing class-aware out-of-sample embeddings that preserve supervised geometric structure. This naturally involves additional computation, such as the generation of random forest-based proximity matrices.
>
> Since UMAP does not use label information during training, it addresses a fundamentally different problem. As such, runtime comparisons between UMAP and RF-AE are not meaningful, as the learning objectives and computational pathways are not aligned. A more appropriate comparison should be made with supervised methods such as CE, SSNP, NCA, and supervised kernel extensions of PHATE and UMAP.
>
> To provide context, we include training and inference time estimates (in seconds) across datasets in the attached tables. Based on average runtime over 10 repetitions on benchmark datasets such as Sign MNIST (Table R4.1) and OrganC MNIST (Table R4.2), RF-AE falls within the same order of magnitude (OoM, i.e. power of ten) as P-SUMAP, NCA, and RF-PHATE during training. It is approximately one OoM slower than CE and SSNP, and about two OoM faster than RF-PHATE’s default kernel extension. At inference time, RF-AE is one OoM slower than P-SUMAP but significantly faster than RF-PHATE.
>
> RF-AE remains scalable across large datasets. Memory usage was not a limiting factor in any of our experiments, and the method benefits from prototype selection and batch-wise processing to keep both computation and memory demand manageable.
>
> In the current implementation, the most time-consuming step is the computation of the RF-GAP proximity matrix. We are actively working on accelerating this step using GPU, which we expect will substantially reduce training time.
>
>
> **Table R4.1: Average training and test times on Sign MNIST, in seconds, across 10 repetitions.**
> |Model|Training Time|Test Time|
> |--------------------|--------------------------------------------|-----------------------------------------|
> | RF-PHATE| $5.69 \times 10^2 \pm 7.74 \times 10^1$| $2.92 \times 10^2 \pm 2.49 \times 10^2$ |
> |RF-AE ($N^{*}=10\\%$)| $6.16 \times 10^2 \pm 3.92 \times 10^1$| $8.45 \times 10^0 \pm 4.27 \times 10^{-1}$ |
> |RF-AE ($N^{*}=2\\%$)| $5.88 \times 10^2 \pm 6.37 \times 10^1$| $1.93 \times 10^0 \pm 1.72 \times 10^{-1}$ |
> |PACMAP| $6.58 \times 10^0 \pm 6.67 \times 10^{-1}$ | $5.24 \times 10^{-1} \pm 4.60 \times 10^{-2}$ |
> |P-SUMAP| $3.62 \times 10^2 \pm 3.87 \times 10^1$| $1.09 \times 10^{-1} \pm 7.00 \times 10^{-3}$ |
> |AE| $2.06 \times 10^1 \pm 1.25 \times 10^0$| $1.90 \times 10^{-2} \pm 2.00 \times 10^{-3}$ |
> |P-TSNE| $1.91 \times 10^1 \pm 7.97 \times 10^{-1}$ | $1.80 \times 10^{-2} \pm 2.00 \times 10^{-3}$ |
> |SSNP| $2.95 \times 10^1 \pm 1.48 \times 10^0$    | $1.80 \times 10^{-2} \pm 2.00 \times 10^{-3}$ |
> |P-UMAP| $1.87 \times 10^1 \pm 8.03 \times 10^{-1}$ | $1.80 \times 10^{-2} \pm 3.00 \times 10^{-3}$ |
> |CE| $5.76 \times 10^1 \pm 3.86 \times 10^0$| $1.70 \times 10^{-2} \pm 1.00 \times 10^{-3}$ |
> |CEBRA| $2.59 \times 10^0 \pm 3.18 \times 10^{-1}$ | $1.10 \times 10^{-2} \pm 7.00 \times 10^{-3}$ |
> |PLS-DA| $2.73 \times 10^{-1} \pm 7.00 \times 10^{-2}$ | $1.10 \times 10^{-2} \pm 3.00 \times 10^{-3}$ |
> |PCA| $3.41 \times 10^{-1} \pm 6.20 \times 10^{-2}$ | $7.00 \times 10^{-3} \pm 1.00 \times 10^{-3}$ |
> |NCA| $3.45 \times 10^2 \pm 2.10 \times 10^2$| $4.00 \times 10^{-3} \pm 1.00 \times 10^{-3}$ |
> |SPCA| $5.62 \times 10^1 \pm 5.65 \times 10^0$| $3.00 \times 10^{-3} \pm 1.00 \times 10^{-3}$ |
>
> **Table R4.2: Average training and test times on OrganC MNIST, in seconds, across 10 repetitions.**
> | Model| Training Time|Test Time|
> |-------------------|------------------------------------|------------------------------------|
> | RF-PHATE| $5.87 \times 10^2 \pm 7.67 \times 10^1$ | $5.05 \times 10^2 \pm 7.16 \times 10^1$ |
> | RF-AE ($N^{*}=10\\%$)| $7.60 \times 10^2 \pm 2.45 \times 10^1$ | $2.71 \times 10^1 \pm 1.24 \times 10^0$ |
> | RF-AE ($N^{*}=2\\%$)| $6.88 \times 10^2 \pm 2.39 \times 10^1$ | $5.89 \times 10^0 \pm 6.50 \times 10^{-2}$ |
> | PACMAP| $7.16 \times 10^0 \pm 7.32 \times 10^{-1}$ | $1.40 \times 10^0 \pm 1.72 \times 10^{-1}$ |
> | P-SUMAP| $2.95 \times 10^2 \pm 3.48 \times 10^1$ | $1.16 \times 10^{-1} \pm 4.00 \times 10^{-3}$ |
> | CE| $5.29 \times 10^1 \pm 9.62 \times 10^0$ | $4.20 \times 10^{-2} \pm 4.00 \times 10^{-3}$ |
> | AE| $1.92 \times 10^1 \pm 2.84 \times 10^0$ | $4.20 \times 10^{-2} \pm 4.00 \times 10^{-3}$ |
> | SSNP| $2.66 \times 10^1 \pm 4.82 \times 10^0$ | $4.00 \times 10^{-2} \pm 7.00 \times 10^{-3}$ |
> | P-TSNE| $1.71 \times 10^1 \pm 4.68 \times 10^0$ | $3.50 \times 10^{-2} \pm 7.00 \times 10^{-3}$ |
> | P-UMAP| $1.70 \times 10^1 \pm 4.81 \times 10^0$ | $3.30 \times 10^{-2} \pm 6.00 \times 10^{-3}$ |
> | PLS-DA| $2.55 \times 10^{-1} \pm 5.10 \times 10^{-2}$ | $3.10 \times 10^{-2} \pm 6.00 \times 10^{-3}$ |
> | CEBRA| $2.11 \times 10^0 \pm 5.33 \times 10^{-1}$ | $2.60 \times 10^{-2} \pm 6.00 \times 10^{-3}$ |
> | PCA| $3.48 \times 10^{-1} \pm 6.50 \times 10^{-2}$ | $2.40 \times 10^{-2} \pm 5.00 \times 10^{-3}$ |
> | NCA| $3.36 \times 10^2 \pm 9.77 \times 10^1$ | $1.10 \times 10^{-2} \pm 1.00 \times 10^{-3}$ |
> | SPCA| $5.60 \times 10^1 \pm 1.22 \times 10^1$ | $7.00 \times 10^{-3} \pm 1.00 \times 10^{-3}$ |
>
>
>
>
> **Re M1.** In our work, we define new data as data points that were not part of the original training set. In practice, these points may come from the same distribution as the training data or from a different one, provided they belong to the same domain. Labels are not required for new data, as all trained models in our comparison—including RF-AE—can generate out-of-sample embeddings without supervision. In our evaluation framework (Section 4), we artificially introduced new data through test splits. We used predefined training/test splits when available; otherwise, we created them using a random 80/20 stratified split to account for class imbalance.
>
>
>
> **Re M2.** While our method belongs to the broader class of dimensionality reduction techniques, our primary focus is on projecting data into 2D or 3D to enable interpretable insights. We believe the term data visualization more accurately reflects this goal, as the purpose of the embedding is to facilitate direct understanding and subsequent annotation of the data. This wording is also typically used in similar papers that perform dimensionality reduction for data visualization.
>
> **References**
>
> [1] Rhodes et al. Geometry- and accuracy-preserving random forest
> proximities. IEEE Transactions on Pattern Analysis and Machine Intelligence, pages 1–13, 2023.

---

> > ### Comment · Reviewer_mQiQ · 2025-08-03
> > **Thank you for the detailed responses!**
> >
> > Thank you for the detailed responses! Table R4.1 and R4.2 are very helpful! I recommend adding them as bar charts in the camera-ready version.
> >
> > Most of my questions have been answered. Given that my original rating was already positive, I will be maintaining it.

---

### Official Review · Reviewer_RUnf · 2025-06-30

**Clarity:** 3
**Significance:** 4
**Originality:** 3
**Rating:** 5
**Confidence:** 3

**Summary:**

This paper introduces a random forest inspired autoencoder (RF-AE), a supervised dimensionality reduction method that combines the "geometric structure" derived from Random Forest proximity kernels (RF-GAP) with the parametric capacity of autoencoders. The proposed framework enables out-of-sample embedding via a geometry-regularized encoder trained to preserve the label-informed structure captured by RF-GAP or RF-PHATE. The approach integrates a novel structural importance alignment metric to assess how well learned embeddings retain class-relevant features. A prototype-based approximation is also proposed to reduce input dimensionality and improve scalability.

**Questions:**

1. The scalability of this framework, as mentioned above.

2. Generalization Beyond RF-PHATE. Have the authors tested RF-AE with other embeddings? Empirical results would substantiate the generality claim.

3. Impact of forest configuration: how sensitive is it to the configuration of RF, such as number of trees, max depth?

**Ethical Concerns:**

["NO or VERY MINOR ethics concerns only"]

**Final Justification:**

This paper, together with the rebuttal results, is well written and sufficiently proves the effectiveness of the proposed method. My concerns about the scalability, computation cost, and extensibility are resolved.

**Limitations:**

Rebuttal content resolves my questions very well.

**Paper Formatting Concerns:**

-

**Quality:**

3

**Strengths And Weaknesses:**

S1: The paper is well-motivated to combine RF-derived proximity with AE training to address the shortcoming of out-of-sample mapping problem of RFs. It enables efficient parametric encoding and retain the supervised neighborhood structure.

S2: It shows strong empirical performance in preserving class-relevant information across various datasets.

S3: The paper is clearly written, well-organized, and provides sufficient implementation detail.

W1: Despite the prototype approximation, the full proximity matrix is initially computed using all training points (L177). This may be prohibitively costly for large datasets. It would be better if more statistics and explanations related to scalability and computational cost could be discussed.

W2: Although it is mentioned that this framework is agnostic to the kernel method. However, only results based on RF-PHATE are included, and it would be better to demonstrate that this framework works with other kernels. How does the selection of the kernel affect the quality of the final representation?

---

> ### Author Rebuttal · Authors · 2025-07-30
>
> We would like to thank you for the positive review and insightful comments. Here are our detailed answers to your  questions.
>
> **Re Q1.** Thank you for this valuable observation. It is true that in our current implementation, the full random forest proximity matrix is computed before selecting prototypes. However, the matrix is not stored in memory permanently. It is held temporarily for processing and then released, so memory usage remains controlled. In practice, memory was not a limiting factor in any of our experiments.
>
> We agree that for very large datasets, the pairwise proximity computation can still be time-consuming. To improve scalability, we are actively working on accelerating this step through GPU-based implementation. This optimization is currently in progress, and we are confident that our GPU-based implementation will be completed by the time the camera-ready submission is due. Unfortunately, per NeurIPS guidelines, external links (e.g., to updated code) are not allowed during the rebuttal period.
>
> To provide a clearer view of computational cost, we include training and test time comparisons in Table R3.1 (Sign MNIST) and Table R3.2 (OrganC MNIST), in seconds, across 10 repetitions. RF-AE falls within the same order of magnitude (OoM, i.e. power of ten) as RF-PHATE, P-SUMAP, and NCA during training, and is approximately one OoM slower than CE and SSNP. At inference time, RF-AE is roughly two OoM faster than RF-PHATE’s default kernel extension and about one OoM slower than P-SUMAP.
>
>
>
> **Table R3.1: Average training and test times on Sign MNIST, in seconds, across 10 repetitions.**
> |Model|Training Time|Test Time|
> |--------------------|--------------------------------------------|-----------------------------------------|
> | RF-PHATE| $5.69 \times 10^2 \pm 7.74 \times 10^1$| $2.92 \times 10^2 \pm 2.49 \times 10^2$ |
> |RF-AE ($N^{*}=10\\%$)| $6.16 \times 10^2 \pm 3.92 \times 10^1$| $8.45 \times 10^0 \pm 4.27 \times 10^{-1}$ |
> |RF-AE ($N^{*}=2\\%$)| $5.88 \times 10^2 \pm 6.37 \times 10^1$| $1.93 \times 10^0 \pm 1.72 \times 10^{-1}$ |
> |PACMAP| $6.58 \times 10^0 \pm 6.67 \times 10^{-1}$ | $5.24 \times 10^{-1} \pm 4.60 \times 10^{-2}$ |
> |P-SUMAP| $3.62 \times 10^2 \pm 3.87 \times 10^1$| $1.09 \times 10^{-1} \pm 7.00 \times 10^{-3}$ |
> |AE| $2.06 \times 10^1 \pm 1.25 \times 10^0$| $1.90 \times 10^{-2} \pm 2.00 \times 10^{-3}$ |
> |P-TSNE| $1.91 \times 10^1 \pm 7.97 \times 10^{-1}$ | $1.80 \times 10^{-2} \pm 2.00 \times 10^{-3}$ |
> |SSNP| $2.95 \times 10^1 \pm 1.48 \times 10^0$    | $1.80 \times 10^{-2} \pm 2.00 \times 10^{-3}$ |
> |P-UMAP| $1.87 \times 10^1 \pm 8.03 \times 10^{-1}$ | $1.80 \times 10^{-2} \pm 3.00 \times 10^{-3}$ |
> |CE| $5.76 \times 10^1 \pm 3.86 \times 10^0$| $1.70 \times 10^{-2} \pm 1.00 \times 10^{-3}$ |
> |CEBRA| $2.59 \times 10^0 \pm 3.18 \times 10^{-1}$ | $1.10 \times 10^{-2} \pm 7.00 \times 10^{-3}$ |
> |PLS-DA| $2.73 \times 10^{-1} \pm 7.00 \times 10^{-2}$ | $1.10 \times 10^{-2} \pm 3.00 \times 10^{-3}$ |
> |PCA| $3.41 \times 10^{-1} \pm 6.20 \times 10^{-2}$ | $7.00 \times 10^{-3} \pm 1.00 \times 10^{-3}$ |
> |NCA| $3.45 \times 10^2 \pm 2.10 \times 10^2$| $4.00 \times 10^{-3} \pm 1.00 \times 10^{-3}$ |
> |SPCA| $5.62 \times 10^1 \pm 5.65 \times 10^0$| $3.00 \times 10^{-3} \pm 1.00 \times 10^{-3}$ |
>
> **Table R3.2: Average training and test times on OrganC MNIST, in seconds, across 10 repetitions.**
> | Model| Training Time|Test Time|
> |-------------------|------------------------------------|------------------------------------|
> | RF-PHATE| $5.87 \times 10^2 \pm 7.67 \times 10^1$ | $5.05 \times 10^2 \pm 7.16 \times 10^1$ |
> | RF-AE ($N^{*}=10\\%$)| $7.60 \times 10^2 \pm 2.45 \times 10^1$ | $2.71 \times 10^1 \pm 1.24 \times 10^0$ |
> | RF-AE ($N^{*}=2\\%$)| $6.88 \times 10^2 \pm 2.39 \times 10^1$ | $5.89 \times 10^0 \pm 6.50 \times 10^{-2}$ |
> | PACMAP| $7.16 \times 10^0 \pm 7.32 \times 10^{-1}$ | $1.40 \times 10^0 \pm 1.72 \times 10^{-1}$ |
> | P-SUMAP| $2.95 \times 10^2 \pm 3.48 \times 10^1$ | $1.16 \times 10^{-1} \pm 4.00 \times 10^{-3}$ |
> | CE| $5.29 \times 10^1 \pm 9.62 \times 10^0$ | $4.20 \times 10^{-2} \pm 4.00 \times 10^{-3}$ |
> | AE| $1.92 \times 10^1 \pm 2.84 \times 10^0$ | $4.20 \times 10^{-2} \pm 4.00 \times 10^{-3}$ |
> | SSNP| $2.66 \times 10^1 \pm 4.82 \times 10^0$ | $4.00 \times 10^{-2} \pm 7.00 \times 10^{-3}$ |
> | P-TSNE| $1.71 \times 10^1 \pm 4.68 \times 10^0$ | $3.50 \times 10^{-2} \pm 7.00 \times 10^{-3}$ |
> | P-UMAP| $1.70 \times 10^1 \pm 4.81 \times 10^0$ | $3.30 \times 10^{-2} \pm 6.00 \times 10^{-3}$ |
> | PLS-DA| $2.55 \times 10^{-1} \pm 5.10 \times 10^{-2}$ | $3.10 \times 10^{-2} \pm 6.00 \times 10^{-3}$ |
> | CEBRA| $2.11 \times 10^0 \pm 5.33 \times 10^{-1}$ | $2.60 \times 10^{-2} \pm 6.00 \times 10^{-3}$ |
> | PCA| $3.48 \times 10^{-1} \pm 6.50 \times 10^{-2}$ | $2.40 \times 10^{-2} \pm 5.00 \times 10^{-3}$ |
> | NCA| $3.36 \times 10^2 \pm 9.77 \times 10^1$ | $1.10 \times 10^{-2} \pm 1.00 \times 10^{-3}$ |
> | SPCA| $5.60 \times 10^1 \pm 1.22 \times 10^1$ | $7.00 \times 10^{-3} \pm 1.00 \times 10^{-3}$ |
>
>
>
>
>
> **Re Q2.** While RF-AE can incorporate any geometric constraint, RF-PHATE is the core focus of our paper, as it ensures that visualizations are aligned with the underlying supervised data geometry. This choice is motivated by Rhodes \& Aumon et al. [1], who empirically demonstrated that RF-PHATE outperformed alternatives.
>
> However, to demonstrate the generality of  geometry-regularized RF-AE network, we provide in Table R3.3 ablation results on Sign MNIST and OrganC MNIST using three alternative geometric constraints: UMAP, Supervised UMAP (SUMAP), and RF-UMAP (i.e., UMAP applied to RF-GAP proximities). We set our default hyperparameters $\lambda=0.01,N^*=0.1$. RF-AE (RF-UMAP) achieved the best overall performance, followed by RF-AE with SUMAP and UMAP constraints. While RF-AE (RF-UMAP) was competitive with RF-AE (RF-PHATE) on OrganC MNIST, it largely underperformed on Sign MNIST. These results suggest that RF-AE is especially effective when paired with RF-based kernel methods—particularly RF-PHATE—which already capture the underlying RF geometry. In such cases, the geometric and reconstruction objectives are well-aligned, enabling more effective multi-task learning. Qualitatively, RF-AE (RF-UMAP) still fragments same-class clusters on the Sign MNIST dataset, though less severely than UMAP and SUMAP. This supports the idea that RF-PHATE more effectively captures both local and global supervised structure through diffusion.
>
> In summary, RF-PHATE is a strong default regularizer for RF-AE overall, though alternative RF-based regularizers like RF-UMAP may offer valuable refinements in specific scenarios. We will include these additional quantitative and qualitative experiments in the Appendix.
>
> **Table R3.3: SIA and kNN accuracies of RF-AE using three different geometric constraints on the Sign MNIST and OrganC MNIST datasets. Scores are averaged across 10 repetitions (mean ± std).**
>
> Dataset | Model         | QNX            | Trust          | Spear          | Pearson        | kNN acc.       |
> |---------------|---------------|----------------|----------------|----------------|----------------|----------------|
> | Sign MNIST | RF-AE (RF-UMAP) | 0.756 ± 0.013 | 0.743 ± 0.015 | 0.505 ± 0.217 | 0.478 ± 0.233 | 0.945 ± 0.010 |
> | | RF-AE (SUMAP)   | 0.692 ± 0.010 | 0.586 ± 0.022 | 0.453 ± 0.075 | 0.400 ± 0.101 | 0.845 ± 0.014 |
> | | RF-AE (UMAP)    | 0.669 ± 0.013 | 0.546 ± 0.018 | 0.311 ± 0.039 | 0.225 ± 0.063 | 0.743 ± 0.020 |
> | OrganC MNIST | RF-AE (RF-UMAP)   | 0.893 ± 0.007    | 0.932 ± 0.005    | 0.934 ± 0.007    | 0.931 ± 0.006    | 0.720 ± 0.009    |
> | | RF-AE (SUMAP)     | 0.881 ± 0.008    | 0.905 ± 0.008    | 0.883 ± 0.008    | 0.872 ± 0.007    | 0.691 ± 0.014    |
> | | RF-AE (UMAP)      | 0.887 ± 0.007    | 0.905 ± 0.006    | 0.867 ± 0.006    | 0.865 ± 0.006    | 0.559 ± 0.025    |
>
>
>
> **Re Q3.** Rhodes et al. [2] demonstrated that the geometry captured by RF-GAP proximities remains stable across varying numbers of trees and minimum node sizes—outperforming other RF-derived kernels in robustness. Moreover, RF-PHATE itself is largely insensitive to its primary hyperparameter, the diffusion time $t$, as shown by Moon et al. [3]. Thus, we did not consider it necessary to perform ablation studies on these secondary hyperparameters, but we are open to doing so if requested.
>
> **References**
>
> [1] Rhodes, Aumon, et al. Gaining biological insights through supervised data visualization. bioRxiv,
> 2024.
>
> [2] Rhodes et al. Geometry- and accuracy-preserving random forest
> proximities. IEEE Transactions on Pattern Analysis and Machine Intelligence, pages 1–13, 2023.
>
> [3] Moon et al. Visualizing structure and transitions in high-dimensional biological data. Nat. Biotechnol.,
> 37(12):1482–1492, Dec 2019.

---

### Official Review · Reviewer_vZZA · 2025-07-02

**Clarity:** 1
**Significance:** 2
**Originality:** 1
**Rating:** 4
**Confidence:** 2

**Summary:**

The paper proposes to build on top of a recent paper (RF-PHATE) to derive a framework for out of sample kernel based visualization. The paper provides a few empirical validations showing more interpretable visualizations.

**Questions:**

Please see above.

**Ethical Concerns:**

["NO or VERY MINOR ethics concerns only"]

**Final Justification:**

Thank you for your answers, please see my last comment addressed to the authors

**Limitations:**

Please see above.

**Quality:**

2

**Strengths And Weaknesses:**

The entire motivation of the paper is unclear to me. Nonlinear visualization of data for out-of-sample has been tackled a long time ago by learning a parametric embedding model of the data and then doing a non-kernel based visualization based on those embeddings (this has been true for decades, even prior Deep Learning). There are also numerous specific papers looking at that problem, such as
- Out-of-Sample Extensions for LLE, Isomap, MDS, Eigenmaps, and Spectral Clustering, Yoshua Bengio, Jean-Francois Paiement, Pascal Vincent Olivier Delalleau, Nicolas Le Roux and Marie Ouimet
- Supervised Deep Feature Embedding With Handcrafted Feature, Shichao Kan; Yigang Cen; Zhihai He; Zhi Zhang; Linna Zhang; Yanhong Wang

While it is possible that I missed the underlying benefits of the method, I believe that the abstract and introduction/background do not provide enough context and motivation which needs to be addressed.

---

> ### Author Rebuttal · Authors · 2025-07-30
>
> We appreciate the reviewer highlighting this important line of work. While the out-of-sample extension problem has indeed been extensively studied—including in classical kernel methods such as Nyström and geometric harmonics—our work addresses a distinct and underexplored setting: *supervised* out-of-sample visualization, where the goal is to preserve both the local structure of the data and class-aware separability in the embedding space. The classic work of Bengio et al. [1] laid important groundwork for out-of-sample extensions using linear mappings for LLE, Isomap, MDS, and spectral methods. However, these approaches typically rely on unconstrained least-squares minimization and are restricted to linear or kernel-based mappings, making them sensitive to the quality of the training data and often inadequate for capturing complex manifold structure in a way that generalizes effectively to new inputs [2,3].
>
> The manifold extension problem remains an active research area, with recent developments such as parametric $t$-SNE [4], parametric UMAP [5], and several geometry-regularized or geometry-aware autoencoders [6--9]. These methods offer more flexible parametric mappings, but most focus either on unsupervised structure or on optimizing embeddings solely for predictive performance, without preserving label-informed geometry that generalizes to new data points.
>
> RF-AE is designed to address this gap by learning an embedding function that preserves both local structure and class separability, based on Random Forest proximities. Unlike supervised methods such as RF-PHATE, which capture label-aware structure but do not offer a learned mapping, RF-AE provides a parametric, interpretable, and generalizable embedding function. Neural methods such as autoencoders can learn mappings, but often lack a strong supervision signal tied to relational structure between points.
>
> As shown in Table 1 (Section 4), RF-AE consistently ranks highest across 20 datasets on both structure-preserving alignment scores and classification accuracy. These results support our claim that RF-AE improves supervised visualization for out-of-sample data by balancing geometric fidelity and label-based information in a way that prior methods do not fully achieve.
>
> We will revise the introduction to more clearly distinguish our motivation and contributions, emphasizing that RF-AE targets interpretable, supervised embeddings that generalize effectively to new data points.
>
> **References**
>
> [1] Bengio et al. Out-of-sample extensions for lle, isomap, mds, eigenmaps, and spectral
> clustering. NeurIPS, 16, 2003.
>
> [2] Rudi et al. Less is more: Nyström computational regularization. In Advances in Neural Information Processing Systems, volume 28. Curran Associates,
> Inc., 2015.
>
> [3]  Quispe et al. Extreme learning machine for out-of-sample extension in laplacian eigenmaps. Pattern Recognition Letters, 74:68–73, April 2016.
>
> [4] van der Maaten et al. Learning a parametric embedding by preserving local structure. In Proceedings of the Twelfth International Conference on Artificial Intelligence and Statistics, volume 5 of Proceedings of Machine Learning Research, pages 384–391, Hilton Clearwater
> Beach Resort, Clearwater Beach, Florida USA, 16–18 Apr 2009. PMLR.
>
> [5] Sainburg et al. Parametric umap embeddings for representation and semisupervised learning. Neural Computation, 33(11):2881–2907, 10 2021.
>
> [6] A.F. Duque, S. Morin, G. Wolf, and K.R. Moon. Geometry regularized autoencoders. IEEE Transactions
> on Pattern Analysis and Machine Intelligence, 45(6):7381–7394, 2022.
>
> [7] Sun et al. Geometry-aware autoencoders for metric learning and generative modeling on data manifolds. ICML 2024 Workshop on Geometry-grounded Representation Learning and Generative Modeling. 2024.
>
> [8] Lee et al. Neighborhood reconstructing autoencoders. In Advances in Neural Information Processing Systems, volume 34, page 536–546. Curran Associates, Inc.,
> 2021.
>
> [9] Nazari et al. Geometric autoencoders - what you see is what you decode. In Proceedings of the 40th International Conference on Machine Learning, volume
> 202 of Proceedings of Machine Learning Research, pages 25834–25857. PMLR, 23–29 Jul 2023.

---

### Official Review · Reviewer_RWaH · 2025-07-03

**Clarity:** 3
**Significance:** 3
**Originality:** 3
**Rating:** 5
**Confidence:** 3

**Summary:**

The paper proposes Random Forest Autoencoders (RF-AE), a new framework for supervised representation learning, especially for out-of-sample (OOS) visualization.
RF-AE integrates the geometry and supervision strengths of RF-PHATE with the flexibility and scalability of autoencoders.
Instead of reconstructing the raw input, RF-AE reconstructs RF-GAP proximities, a supervised similarity metric derived from random forests, which allows it to learn embeddings aligned with the supervised structure of the data.
A geometric regularization term ensures alignment between the latent representation and a precomputed RF-PHATE embedding.
The authors also introduce a prototype selection mechanism to reduce computational costs and memory usage by selecting representative training points via class-wise k-medoids clustering on RF-GAP dissimilarities.
RF-AE consistently outperforms baseline parametric and nonparametric methods in terms of both classification accuracy and a novel Structural Importance Alignment (SIA) metric, which measures how well the embedding reflects the classification-relevant structure of the input features.
RF-AE achieves strong trade-offs between local/global structure preservation and class separability, with robust performance across various hyperparameter settings.

**Questions:**

1. While the authors acknowledge the computational cost of RF-AE, there is no concrete study and comparison on it, except the running time by different amount of prototypes. There should be training/inference time (and memory consumption) comparison with the other baselines compared across datasets, for a better understanding for practical considerations.
2. How sensitive is the SIA metric to the choice of classifiers and perturbation schemes? Could the authors provide ablation results or sensitivity analyses with different classifiers or correlation thresholds? The authors may consider reporting SIA variation across: alternative classifiers or non-ensembled versions, different perturbation strategies, and/or datasets with weak label–feature correlation.
3. Can we utilize unlabeled data also for training RF-AE, e.g., in a semi-supervised way? Unsupervised methods of course can handle all available data without using labels, and methods like UMAP, semi-supervised variants of t-SNE [1], ViVA [2], etc., can learn from both unlabeled and labeled data. In a realistic scenario, learning from labeled data only may reduce data efficiency.

[1] Serna-Serna, Walter, et al. "Semi-supervised t-SNE with multi-scale neighborhood preservation." Neurocomputing 550 (2023): 126496.

[2] An, Sungtae, Shenda Hong, and Jimeng Sun. "Viva: semi-supervised visualization via variational autoencoders." 2020 IEEE International Conference on Data Mining (ICDM). IEEE, 2020.

**Ethical Concerns:**

["NO or VERY MINOR ethics concerns only"]

**Final Justification:**

This paper proposes a novel method for supervised dimensionality reduction, especially for data visualization.
The paper is technically solid and well-supported by experimental results.
Most initial concerns on scalability (running time), semi-supervised learning capability, and potential limitations were addressed during the rebuttal period.

**Limitations:**

The authors have done a good job acknowledging technical limitations, particularly about scalability due to the computational cost of RF-GAP proximities.
However, they do not sufficiently address potential negative societal impacts of their method, especially considering its applicability in high-stakes domains like biomedical and healthcare.
It would be very helpful if the authors could discuss some potential pitfalls, for example (not must):
- Risks in biomedical or clinical data visualization, such as over-interpretation of embeddings, model bias from supervised proximities, or misuse of visualizations in decision-making.
- How the complexity of the RF-GAP kernel and the autoencoder might make the resulting embeddings even harder to interpret, especially by non-technical stakeholders.
- What if the labels RF-GAP relies on encode social or demographic biases? Could this lead to skewed embeddings?

Addressing those kinds of concerns, even briefly, would demonstrate awareness of broader impact and strengthen the paper’s maturity.

**Quality:**

3

**Strengths And Weaknesses:**

**Strengths**
- The proposed RF-AE framework combines random forest-derived (RF-GAP) proximities with an autoencoder-based architecture. Reconstructing RF-GAP proximities, rather than original input vectors, within the autoencoder is a meaningful and novel departure from standard AE objectives.
- The methodological foundation is strong, integrating insights from manifold learning, kernel regression, and autoencoder design in a coherent way. Key innovations include geometric loss regularization based on RF-PHATE embeddings and a prototype selection mechanism to enhance scalability.
- The geometric loss, guided by RF-PHATE embeddings, ensures alignment between the supervised structure and the learned latent space, helping mitigate overfitting or trivial embedding solutions.
- The Structural Importance Alignment (SIA) metric fills an important evaluation gap in supervised visualization by quantifying the extent to which embeddings preserve classification-relevant feature structure. This is a valuable and well-motivated contribution.
- RF-AE ranks among the top across both local/global SIA and classification accuracy benchmarks. Ablation studies further demonstrate robustness to hyperparameter choices such as the geometric constraint weight ($\lambda$) and number of prototypes.

**Weaknesses**
- As the authors acknowledge, a key limitation is the computational cost of computing RF-GAP proximities, which scales poorly with dataset size. Although prototype selection via class-wise k-medoids helps mitigate this, it still requires full computation of the RF-GAP proximity matrix which is posing a barrier for large-scale applications.
- There are no concrete runtime or memory benchmarks demonstrating feasibility on large-scale datasets, although timing for the proposed method on a single dataset with varying prototype counts is reported.
- The generalizability of the SIA metric may depend on the specific ensemble of classifiers and the perturbation heuristics used, which could introduce variance or bias across different data modalities or domains.

---

> ### Author Rebuttal · Authors · 2025-07-30
>
> Thank you for the valuable feedback and overall positive review. Below, we provide a point-by-point
> response to the concerns you raised.
>
> **Re Q1.** In the revised version, we will include a training and inference time comparison in the Appendix. While RF-AE incurs additional training time due to the computation of the random forest proximity, we note that it significantly improves out-of-sample supervised embeddings (see quantitative results in Table 1), which we believe makes the additional cost worthwhile in many practical scenarios.
>
> All experiments were conducted on a shared computing cluster, where compute nodes are allocated dynamically. As a result, runtimes may vary depending on system load and available resources. To give a rough estimate of RF-AE’s runtime relative to other baselines, we report the training and test times on the Sign MNIST (Table R1.1) and OrganC MNIST (Table R1.2) datasets. RF-AE falls within the same order of magnitude (OoM) as RF-PHATE, P-SUMAP, and NCA during training, and one OoM slower than CE and SSNP. At inference time, RF-AE is approximately 2 OoM faster than RF-PHATE, and one OoM slower than P-SUMAP.
> Memory usage did not pose a constraint in any of our experiments. RF-AE scales well across datasets without requiring extensive memory resources, and our design choices, including our prototype selection, help keep computations and memory demand manageable.
>
> Furthermore, we are actively optimizing our RF-GAP implementation by migrating proximity matrix computation to the GPU (in progress) to improve efficiency. These updates will be reflected in our GitHub repository once completed.
>
> **Table R1.1: Average training and test times on Sign MNIST, in seconds, across 10 repetitions.**
> |Model|Training Time|Test Time|
> |--------------------|--------------------------------------------|-----------------------------------------|
> | RF-PHATE| $5.69 \times 10^2 \pm 7.74 \times 10^1$| $2.92 \times 10^2 \pm 2.49 \times 10^2$ |
> |RF-AE ($N^{*}=10\\%$)| $6.16 \times 10^2 \pm 3.92 \times 10^1$| $8.45 \times 10^0 \pm 4.27 \times 10^{-1}$ |
> |RF-AE ($N^{*}=2\\%$)| $5.88 \times 10^2 \pm 6.37 \times 10^1$| $1.93 \times 10^0 \pm 1.72 \times 10^{-1}$ |
> |PACMAP| $6.58 \times 10^0 \pm 6.67 \times 10^{-1}$ | $5.24 \times 10^{-1} \pm 4.60 \times 10^{-2}$ |
> |P-SUMAP| $3.62 \times 10^2 \pm 3.87 \times 10^1$| $1.09 \times 10^{-1} \pm 7.00 \times 10^{-3}$ |
> |AE| $2.06 \times 10^1 \pm 1.25 \times 10^0$| $1.90 \times 10^{-2} \pm 2.00 \times 10^{-3}$ |
> |P-TSNE| $1.91 \times 10^1 \pm 7.97 \times 10^{-1}$ | $1.80 \times 10^{-2} \pm 2.00 \times 10^{-3}$ |
> |SSNP| $2.95 \times 10^1 \pm 1.48 \times 10^0$    | $1.80 \times 10^{-2} \pm 2.00 \times 10^{-3}$ |
> |P-UMAP| $1.87 \times 10^1 \pm 8.03 \times 10^{-1}$ | $1.80 \times 10^{-2} \pm 3.00 \times 10^{-3}$ |
> |CE| $5.76 \times 10^1 \pm 3.86 \times 10^0$| $1.70 \times 10^{-2} \pm 1.00 \times 10^{-3}$ |
> |CEBRA| $2.59 \times 10^0 \pm 3.18 \times 10^{-1}$ | $1.10 \times 10^{-2} \pm 7.00 \times 10^{-3}$ |
> |PLS-DA| $2.73 \times 10^{-1} \pm 7.00 \times 10^{-2}$ | $1.10 \times 10^{-2} \pm 3.00 \times 10^{-3}$ |
> |PCA| $3.41 \times 10^{-1} \pm 6.20 \times 10^{-2}$ | $7.00 \times 10^{-3} \pm 1.00 \times 10^{-3}$ |
> |NCA| $3.45 \times 10^2 \pm 2.10 \times 10^2$| $4.00 \times 10^{-3} \pm 1.00 \times 10^{-3}$ |
> |SPCA| $5.62 \times 10^1 \pm 5.65 \times 10^0$| $3.00 \times 10^{-3} \pm 1.00 \times 10^{-3}$ |
>
> **Table R1.2: Average training and test times on OrganC MNIST, in seconds, across 10 repetitions.**
> | Model| Training Time|Test Time|
> |-------------------|------------------------------------|------------------------------------|
> | RF-PHATE| $5.87 \times 10^2 \pm 7.67 \times 10^1$ | $5.05 \times 10^2 \pm 7.16 \times 10^1$ |
> | RF-AE ($N^{*}=10\\%$)| $7.60 \times 10^2 \pm 2.45 \times 10^1$ | $2.71 \times 10^1 \pm 1.24 \times 10^0$ |
> | RF-AE ($N^{*}=2\\%$)| $6.88 \times 10^2 \pm 2.39 \times 10^1$ | $5.89 \times 10^0 \pm 6.50 \times 10^{-2}$ |
> | PACMAP| $7.16 \times 10^0 \pm 7.32 \times 10^{-1}$ | $1.40 \times 10^0 \pm 1.72 \times 10^{-1}$ |
> | P-SUMAP| $2.95 \times 10^2 \pm 3.48 \times 10^1$ | $1.16 \times 10^{-1} \pm 4.00 \times 10^{-3}$ |
> | CE| $5.29 \times 10^1 \pm 9.62 \times 10^0$ | $4.20 \times 10^{-2} \pm 4.00 \times 10^{-3}$ |
> | AE| $1.92 \times 10^1 \pm 2.84 \times 10^0$ | $4.20 \times 10^{-2} \pm 4.00 \times 10^{-3}$ |
> | SSNP| $2.66 \times 10^1 \pm 4.82 \times 10^0$ | $4.00 \times 10^{-2} \pm 7.00 \times 10^{-3}$ |
> | P-TSNE| $1.71 \times 10^1 \pm 4.68 \times 10^0$ | $3.50 \times 10^{-2} \pm 7.00 \times 10^{-3}$ |
> | P-UMAP| $1.70 \times 10^1 \pm 4.81 \times 10^0$ | $3.30 \times 10^{-2} \pm 6.00 \times 10^{-3}$ |
> | PLS-DA| $2.55 \times 10^{-1} \pm 5.10 \times 10^{-2}$ | $3.10 \times 10^{-2} \pm 6.00 \times 10^{-3}$ |
> | CEBRA| $2.11 \times 10^0 \pm 5.33 \times 10^{-1}$ | $2.60 \times 10^{-2} \pm 6.00 \times 10^{-3}$ |
> | PCA| $3.48 \times 10^{-1} \pm 6.50 \times 10^{-2}$ | $2.40 \times 10^{-2} \pm 5.00 \times 10^{-3}$ |
> | NCA| $3.36 \times 10^2 \pm 9.77 \times 10^1$ | $1.10 \times 10^{-2} \pm 1.00 \times 10^{-3}$ |
> | SPCA| $5.60 \times 10^1 \pm 1.22 \times 10^1$ | $7.00 \times 10^{-3} \pm 1.00 \times 10^{-3}$ |
>
> **Re Q2.** We analyzed the SIA performance of all models under different classification importance strategies, as detailed in Appendix I. In addition to our ensemble approach, we derived feature importances from a standalone $k$-NN classifier, as well as from an aggregated strategy where importances were computed separately from each model ($k$-NN, SVM, MLP) and then combined. We observed that RF-AE consistently ranked at the top across all metrics, regardless of the feature importance strategy. This suggests that RF-AE is able to preserve meaningful structure in a more general sense—not just from the perspective of a single classifier. As for a sensitivity analysis of SIA under different perturbation strategies, we are not aware of other correlation-aware schemes beyond our own, which is inspired by Kaneko [1]. That said, we are open to exploring additional strategies in future work. Finally, although not explicitly stated in the manuscript, feature importances were averaged over 10 random sampling procedures to ensure robustness. We will clarify these points in the final paper.
>
> **Re Q3.** Yes: RF-AE supports partially labeled datasets, making it a compelling approach in semi-supervised settings compared to strictly class-conditional methods. As noted in Rhodes \& Aumon et al. [2]: “ (...) a trained random forest can define similarities to out-of-sample observations, supporting semi-supervised learning (...) ” This is further elaborated in our manuscript (L155): “Note that this definition (of RF-GAP) naturally extends to OOS observations (...) which can be treated as out-of-bag for all trees.” Thus, on the one hand, assuming $N_L$ labeled points and $N_U$ unlabeled points, for a total training size of $N = N_L + N_U$, we treat unlabeled training samples as “out-of-sample” and compute $N$ proximity vectors of size $N_L$, which are used as input to train the RF-AE network. On the other hand, to generate RF-PHATE embeddings for the full training set, we can rely on the Landmark PHATE algorithm proposed by Moon et al. [3]. We first construct the RF-GAP kernel matrix of size $N_{L} \times N_{L}$ between labeled landmarks and compute their embeddings with PHATE. Then, we project the $N_{U}$ remaining unlabeled points with the linear landmark extension using their RF-GAP proximities to the labeled points (the landmarks), as in L118-L122. This provides all the necessary components to train our regularized RF-AE using information from both labeled and unlabeled data.  We will add this discussion to the final version of the paper.
>
> **Re Limitations:** While technically involved, the core idea of RF-AE—embedding data in 2D—is intuitive. Autoencoders learn low-dimensional representations that capture intrinsic structure. The RF-GAP kernel (Section 3.1) replaces standard Euclidean measures with similarities based on discriminative features, effectively downweighting irrelevant ones. RF-PHATE regularization further denoises the embedding and captures long-range relationships. Combined, these elements let RF-AE organize data based on local and global RF-GAP similarities: nearby points share key features, while distant ones differ meaningfully. We validated this using interpretable metrics (Section 4.1; $k$-NN, SIA metrics) and qualitatively through 2D visualizations showing stronger class-aware structure than competing methods.Thus we do not believe that the results of RF-AE will be substantially more difficult to interpret than other, similar methods.
>
> Still, we agree that supervised 2D visualizations should be interpreted with caution due to potential biases in label assignment. When label assignments reflect social or demographic biases, supervised methods are prone to embedding structural biases, as they aim to discriminate between classes. Another source of label bias arises in highly imbalanced scenarios. In such cases, the Random Forest is biased toward majority classes. As a result, minority classes may appear closer to or farther from other classes than expected, since the features that characterize them may not be adequately captured in the RF-GAP proximities.
> We note that these concerns are not specific to RF-AE but apply broadly to supervised methods. Additionally, RF-AE helps mitigate these effects by avoiding the exaggerated separations often introduced by purely class-conditional methods. These and the other ethical considerations raised by the reviewer will be discussed in a dedicated section of the manuscript.
>
> [1] Kaneko. Cross-validated permutation feature importance considering correlation between
> features. Analytical Science Advances, 2022
>
> [2] Rhodes, Aumon, et al. Gaining biological insights through supervised data visualization. bioRxiv,
> 2024
>
> [3] Moon et al. Visualizing structure and transitions in high-dimensional biological data. Nat.
> Biotechnol., Dec 2019

---

> > ### Comment · Reviewer_RWaH · 2025-08-04
> >
> > Thank you for the detailed answers and experimental results despite the short time.
> > It would be very helpful for readers if the authors include these experimental results about running times and discussion of other aspects (partial label utilization, label bias vulnerability, etc.) in the revised or camera-ready version of the paper, as well as in the code packages.
> > Most my concerns have been addressed and I change my recommendation to accept.

---

### Note · Authors · 2025-08-14

We would like to thank again the reviewers for their meaningful feedback, especially for their engagement in the discussion. We highly value their raised concerns which led us to improve our paper. Here is a summary of our main modifications.

**On computational efficiency.**
RWaH, RUnf, and mQiQ raised concerns about missing runtime comparisons. We added training and test times (Appendix) for RF-AE and all baselines on Sign MNIST and OrganC MNIST. RF-AE trains in the same order of magnitude (OoM) as RF-PHATE, P-SUMAP, and NCA, and is one OoM slower than CE and SSNP. At inference, it is ~2 OoM faster than RF-PHATE and one OoM slower than P-SUMAP. While training is slower due to RF-GAP computation, this is offset by improved out-of-sample embeddings (Table 1). Memory was never a bottleneck, and prototype selection kept usage manageable. We are migrating RF-GAP to GPU, which already reduces training time by 50–70%, with updates to be released on GitHub.

**On novelty and motivation.**
vZZA requested clearer novelty and benefits. While OOS extensions are well studied, most are limited to linear/kernel mappings or focus on unsupervised geometry, overlooking label-informed structure. RF-AE targets the underexplored setting of *supervised* OOS visualization, learning a parametric, interpretable mapping that preserves both local geometry and class separability. Unlike RF-PHATE, which lacks a learned mapping, RF-AE generalizes well to new data while maintaining class-aware structure, as shown by top rankings across 20 datasets on SIA scores and k-NN accuracy (Table 1). We revised our introduction to better explain this supervised setting, why it is different from past OOS work, and what makes RF-AE useful.

**On other geometric constraints.**
RUnf asked about compatibility with other regularizers. We tested UMAP (unsupervised), SUMAP, and RF-UMAP (supervised) in place of RF-PHATE, with quantitative results and 2D visualizations on Sign and OrganC MNIST. Overall, they degraded supervised embedding quality, consistent with prior works showing RF-PHATE’s strength in capturing denoised, label-informed structure. Still, RF-UMAP showed benefits in some cases, suggesting other RF-based regularizers merit exploration.

**On semi-supervised training and ethics.**
We discussed adapting RF-AE for partially labeled data via labeled landmarks, and caution against misleading plots when labels are biased or data are imbalanced—although not specific to RF-AE.

---

### Decision · Program_Chairs · 2025-09-17

**Decision:**

Accept (poster)

**Comment:**

All reviews leaned towards acceptance with two accept [RWaH,RUnf] and two borderline accept [vZZA,mQiQ] recommendations.

The reviewers appreciated many parts of the work:

+ Developing new supervised dimension reduction algorithms was considered a timely topic [mQiQ]
+ The paper was considered well motivated [RUnf]
+ The idea of reconstructing RF-GAP proximities was seen as meaningful and novel [RWaH]; similarly, the method was considered sound and novel by another reviewer [mQiQ]
+ The methodological foundation was considered strong [RWaH]
+ Using the geometric loss in alignment to mitigate overfitting and trivial solutions was appreciated [RWaH]
+ The Structural Importance Alignment (SIA) metric was appreciated as a contribution to evaluation of supervised visualisation [RWaH]
+ The experiments were considered comprehensive [mQiQ]
+ The good empirical performance was appreciated [RWaH,RUnf,mQiQ] and robustness in ablation studies was appreciated [RWaH]
+ The paper was considered clearly written [RUnf] and well-organised and easy to follow [mQiQ]

However, weaknesses were also pointed out, and authors responded to them in the rebuttal stage:

- The motivation and novelty were considered unclear compared to existing work on out-of-sample extensions [vZZA]; authors provided discussion.
- Lack of use of auxiliary/expert labels was criticised [mQiQ]; authors provided discussion arguing their use of labels means expert labels.
- The computational cost was seen as a limitation [RWaH] including especially initial computation of the full prototype matrix [RUnf]; and scalability and computational cost discussion was desired [RUnf]; authors provided some discussion and tables of training and test times.
- Similarly, scalability versus UMAP was a concern [mQiQ]; authors argued UMAP addresses a different problem and that comparisons would not be meaningful; their table of training/test times included one UMAP-related method.
- Showing results only based on RF-PHATE was criticised [RUnf] and showing results with other kernels was desired [RUnf]; authors provided some discussion and results with other geometric constraints.
- Sensitivity to the forest configuration was a concern [RUnf]; authors provided some discussion but not new results.
- Lack of runtime and memory benchmarks for feasibility on large-scale datasets was criticised [RWaH]; authors promised to include a training and inference time comparison and reported some results.
- There was concern the SIA metric might depend on specifics of the classifier ensemble and perturbation heuristics [RWaH]; authors provided some arguments related to this.
- There was a question os using semi-supervised training [RWaH] and authors argued this is possible but did not provide results.
- Addressing negative societal impacts was desired [RWaH]; authors provided some discussion.
- There was concern about the term visualisation versus dimensionality reduction [mQiQ]; authors argued their goal is more accurately described as visualisation.

Ultimately, reviewer [RUnf] considered their concerns addressed, and [RWaH,mQiQ] considered that most of their concerns were addressed; [vZZA] acknowledged the discussion but did not comment in detail whether the concerns were addressed.

Overall, it seems the reviewers appreciate the work, and the majority of concerns were addressed in the discussion. Overall, it seems the work could be ready for presentation at NeurIPS.